# Synthesizing Multimodal Geometry Datasets from Scratch and Enabling Visual Alignment via Plotting Code

**Haobo Lin** [1 *]  **Tianyi Bai** [2 * ‡]  **Chen Chen** [1]  **Jiajun Zhang** [3]  **Bohan Zeng** [4]  **Wentao Zhang** [4]  **Binhang Yuan** [2 †]

## Abstract

Multimodal geometry reasoning requires models to jointly understand visual diagrams and perform structured symbolic inference, yet current vision–language models struggle with complex geometric constructions due to limited training data and weak visual–symbolic alignment. We propose a pipeline for synthesizing complex multimodal geometry problems from scratch and construct a dataset named **GeoCode**, which decouples problem generation into symbolic seed construction, grounded instantiation with verification, and code-based diagram rendering, ensuring consistency across structure, text, reasoning, and images. Leveraging the plotting code provided in GeoCode, we further introduce code prediction as an explicit alignment objective, transforming visual understanding into a supervised structured prediction task. GeoCode exhibits substantially higher structural complexity and reasoning difficulty than existing benchmarks, while maintaining mathematical correctness through multi-stage validation. Extensive experiments show that models trained on GeoCode achieve consistent improvements on multiple geometry benchmarks, demonstrating both the effectiveness of the dataset and the proposed alignment strategy.

## 1. Introduction

Geometry reasoning has long been regarded as a core testbed for multimodal intelligence. Unlike natural image understanding or general visual question answering, geometry problems require precise visual perception, structured symbolic representations, and multi-step logical reasoning to jointly support a verifiable conclusion. (Xu et al., 2025;

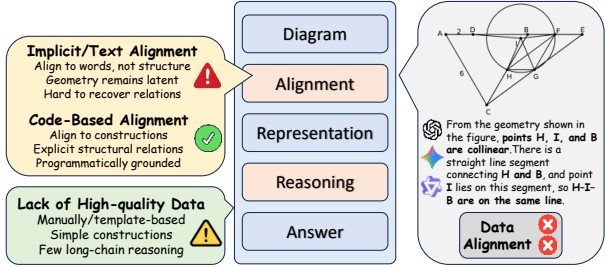

*Figure 1.* **Motivation of our work.** We address the limitations of implicit visual–symbolic alignment and low structural diversity and low problem difficulty in current geometry datasets by synthesizing problems from scratch and supervising models with plotting code for explicit structural grounding.

Ping et al., 2025; Weng et al., 2025; Cho et al., 2025) Recent advances in vision–language models (VLMs)(Li et al., 2022; Liu et al., 2023; Li et al., 2023; OpenAI, 2023; Bai et al., 2023; Wang et al., 2024; Chen et al., 2024) have led to notable progress on basic geometry-related benchmarks(Lu et al., 2021; Chen et al., 2021; Cao & Xiao, 2022; He et al., 2024; Lu et al., 2023; Zhang et al., 2024a). However, when faced with problems involving complex constructions, dense relational constraints, and long-chain reasoning, current models still exhibit substantial limitations, particularly in their ability to recover geometric structure from diagrams and to maintain consistency across intermediate reasoning steps(Lu et al., 2023; Zhang et al., 2024a; Xu et al., 2025).

From a problem-solving perspective, multimodal geometry reasoning is commonly decomposed into two stages: (1) visual–symbolic alignment, which recovers structured geometric entities and relational constraints from diagrams, and (2) symbolic reasoning, which performs deductive inference under joint visual and textual conditions. Recent advances in formal geometry solvers show that, when accurate symbolic representations are available, purely symbolic systems can solve problems approaching International Mathematical Olympiad (IMO) level without any visual input (Trinh et al., 2024; Chervonyi et al., 2025), indicating that long-horizon geometric deduction itself is largely tractable. In contrast, despite recent progress, multimodal models still remain substantially behind symbolic geometry solvers on challenging problems involving complex constructions and dense

---
[*]Equal contribution. [†]Corresponding author. [‡]Project leader. [1]Jilin University, [2]The Hong Kong University of Science and Technology, [3]University of Science and Technology of China, [4]Peking University. Correspondence: biyuan@ust.hk.

*Proceedings of the 43rd International Conference on Machine Learning*, Seoul, South Korea. PMLR 306, 2026. Copyright 2026 by the author(s).

relational constraints. Empirical evaluations on geometry benchmarks reveal persistent failures in recovering accurate geometric structures and maintaining intermediate-step consistency, suggesting that substantial challenges remain in enabling robust multimodal geometry reasoning (Lu et al., 2023; Zhang et al., 2024a; He et al., 2024).

Existing methods attempt to mitigate this gap through improved vision–language alignment (Li et al., 2024; Zhang et al., 2024b), synthetic data generation with reasoning supervision (Gao et al., 2023; Deng et al., 2024; Wang et al., 2025), or solver-driven pipelines that ensure symbolic correctness (Pan et al., 2025; Fu et al., 2025), achieving notable progress on standard benchmarks. In this work, we focus on two central problems that remain unresolved. **First, data.** Existing geometry data are often produced by manual design, template-based augmentation, or solver-driven generation. Solver-based methods are attractive because they provide symbolic correctness, but many of them couple symbolic search with numerical realization by directly extending the solver to support metric quantities. This coupling is difficult: symbolic reasoning operates over a discrete combinatorial space, while numerical instantiation lives in a continuous coordinate space. As a result, practical systems often rely on fixed templates, constrained construction patterns, or hand-designed numerical choices to bridge the two spaces, which limits both structural diversity and metric variation. **Second, alignment.** Most multimodal geometry methods supervise models through final answers, reasoning traces, or natural-language descriptions of diagrams. Such supervision improves task performance, but it does not explicitly require the model to recover the geometric entities, segments, circles, and annotations that determine the visual structure. Consequently, geometric structure recovery remains weakly constrained and largely implicit.

Unlike general multimodal reasoning tasks (Yue et al., 2024), geometry possesses a distinctive advantage: its structures are both symbolically solvable and numerically verifiable via coordinate-based constructions, enabling strict validation of geometric consistency. This property makes it feasible to synthesize geometry problems from scratch with correctness guarantees and to enforce cross-modal consistency among symbolic structures, language, reasoning traces, and rendered diagrams, an opportunity that remains underexploited in existing multimodal geometry datasets.

To address the data problem, we propose a pipeline for synthesizing multimodal geometry problems from scratch and construct a dataset named **GeoCode**. The key idea is to decouple symbolic structure discovery from numerical instantiation and visual rendering. Our pipeline first constructs symbolic geometric seeds via dependency analysis, ensuring that each seed admits non-trivial deductive targets at the symbolic level. Large language models then instantiate these abstract structures with concrete numerical parameters, natural-language problem statements, and reasoning traces, while a separate Coder module generates executable meta code for coordinate-level constructions. Finally, strict semantic validation, coordinate-based geometric verification, and visual quality filtering remove inconsistent or ill-posed samples. As a result, each retained instance maintains cross-modal consistency across symbolic structure, textual description, reasoning process, plotting code, and rendered diagram, without relying on a small set of fixed numerical templates.

To address the alignment problem, we use plotting code as an explicit supervision target. Instead of relying only on final answers, natural-language rationales, or implicit vision–language alignment, plotting-code prediction requires models to reconstruct the full geometric structure from images, including points, segments, circles, annotations, construction dependencies, and spatial relations. This transforms visual understanding into a structured and verifiable prediction task, making geometric perception an explicit component of the learning objective.

Extensive experiments demonstrate that our synthesized dataset exhibits higher structural complexity and reasoning difficulty than existing geometry benchmarks, and that models trained on our data achieve consistent improvements across multiple public geometry datasets, also under out-of-distribution evaluation. Ablation studies further confirm the critical role of plotting-code supervision in strengthening visual–symbolic alignment, showing clear advantages over text-only alignment. We argue that geometric diagrams are inherently complete structural representations, whereas language is unavoidably lossy and ambiguous; therefore, effective geometry reasoning should be grounded in explicit structural representations rather than relying solely on linguistic supervision.

The main contributions of this work are summarized as follows:

- We propose a pipeline to synthesize multimodal geometry problems from scratch, with verification across symbolic structure, language description, reasoning trace, and visual diagram.

- We construct **GeoCode**, an 18K multimodal geometry dataset with diagrams, solutions, and plotting code, while achieving significantly higher difficulty and structural complexity under strict correctness.

- We introduce plotting code as an explicit supervision target for cross-modal alignment, enabling structured learning of geometric relations from visual diagrams.

- We show consistent improvements on multiple public geometry benchmarks, including in out-of-distribution

settings.

## 2. Related Work

**Multimodal Large Language Models and Reasoning** Recent years have witnessed rapid advances in large language models (LLMs), leading to strong performance across a wide range of reasoning tasks. The introduction of GPT-4(Achiam et al., 2023) and subsequent vision–language models(OpenAI, 2023) has further shifted the community's focus toward multimodal understanding(Chen et al., 2024; Wang et al., 2024), where extracting and integrating visual information becomes an essential capability. Data-centric studies further highlight that model capability is strongly shaped by the scale, diversity, and supervision form of multimodal training data(Bai et al., 2024). To enhance reasoning, many works adopt explicit intermediate supervision such as Chain-of-Thought (CoT)(Wei et al., 2022), enabling models to generate multi-step reasoning traces, while recent multi-step visual reasoning methods improve inference through visual token scaling and verification(Bai et al., 2025c). More recently, reinforcement learning based alignment methods, including PPO(Schulman et al., 2017) and GRPO(Shao et al., 2024), have been widely applied to further optimize reasoning behaviors under task-specific objectives or preference feedback(Zhu et al., 2026; Lin et al., 2026; Liang et al., 2026a;b), yielding consistent improvements on complex reasoning benchmarks.

**Multimodal Geometry Problem Solving.** Geometry problem solving is a challenging multimodal task that requires jointly understanding visual diagrams and performing structured symbolic reasoning. With the development of multimodal LLMs, MGPS has become an important testbed for evaluating visual–symbolic reasoning. Existing approaches mainly fall into three categories: alignment-oriented methods such as EAGLE(Li et al., 2024) and MAVIS(Zhang et al., 2024b) that enhance visual perception via staged vision–language alignment, data-centric methods such as R-CoT(Deng et al., 2024), G-LLaVA(Gao et al., 2023), and MathCoder-VL(Wang et al., 2025) that construct synthetic multimodal training data with reasoning traces, and solver-driven generation methods that leverage symbolic geometry solvers such as AlphaGeometry(Trinh et al., 2024), with representative works including GeoGen(Pan et al., 2025) and TrustGeoGen(Fu et al., 2025). However, two fundamental challenges remain. *First*, the lack of large-scale, high-quality geometry datasets with sufficient structural complexity and strict cross-modal consistency limits models' ability to learn long-range geometric dependencies and multi-step reasoning patterns. *Second*, effective visual–symbolic alignment mechanisms are still underexplored, as most methods supervise only final answers or textual reasoning without explicitly constraining geometric structure reconstruction from diagrams.

## 3. Method

In this section, we present a generation pipeline for synthesizing complex multimodal geometry problems from scratch. We first construct symbolic geometric structures and instantiate them with concrete metrics, and then render them into visual diagrams to ensure strict cross-modal consistency in Section 3.1. Building on this pipeline, we further introduce plotting code as an explicit alignment target in Section 3.2, which supervises models to recover geometric structure directly from diagrams.

### 3.1. Generation Pipeline

As illustrated in Fig. 2, our generation pipeline factorizes multimodal geometry problem synthesis into three modular and verifiable stages: (i) symbolic seed generation for constructing relational structures, (ii) grounded instantiation with semantic and geometric verification, and (iii) visualization with textual debiasing. This decomposition disentangles logical structure, metric realization, and perceptual presentation, enabling controllable synthesis while maintaining strict cross-modal alignment.

#### 3.1.1. SEED GENERATION

This stage focuses on constructing symbolic geometric structures without numerical grounding or linguistic formulation. While random sampling and theorem provers are effective at exploring and validating relational configurations, assigning metric values that both satisfy constraints and yield pedagogically meaningful problems is considerably more difficult and error prone. We therefore decouple structural discovery from metric instantiation, and restrict this stage to symbolic reasoning over geometric predicates.

Seed generation starts from random sampling of predicate sets, which specify relations such as incidence, parallelism, perpendicularity, and equality among abstract geometric entities. These sampled predicates form candidate relational graphs, serving as hypotheses in the symbolic space.

**Target Finder via Dependency Construction.** Given a predicate set $S$ and a theorem library $\mathcal{T}$, we apply forward symbolic reasoning to derive all propositions that are logically implied by $S$:

$$\mathcal{D}(S) = \{\, p \mid S \vdash_{\mathcal{T}} p \,\}.$$

AlphaGeometry is used as the reasoning engine to construct dependency graphs, where nodes correspond to predicates and edges encode proof dependencies. Each valid reasoning target is associated with a proof graph consisting of premises, intermediate steps, and a final conclusion. This process determines whether the sampled structure admits non-trivial deductive chains rather than isolated or redundant constraints.

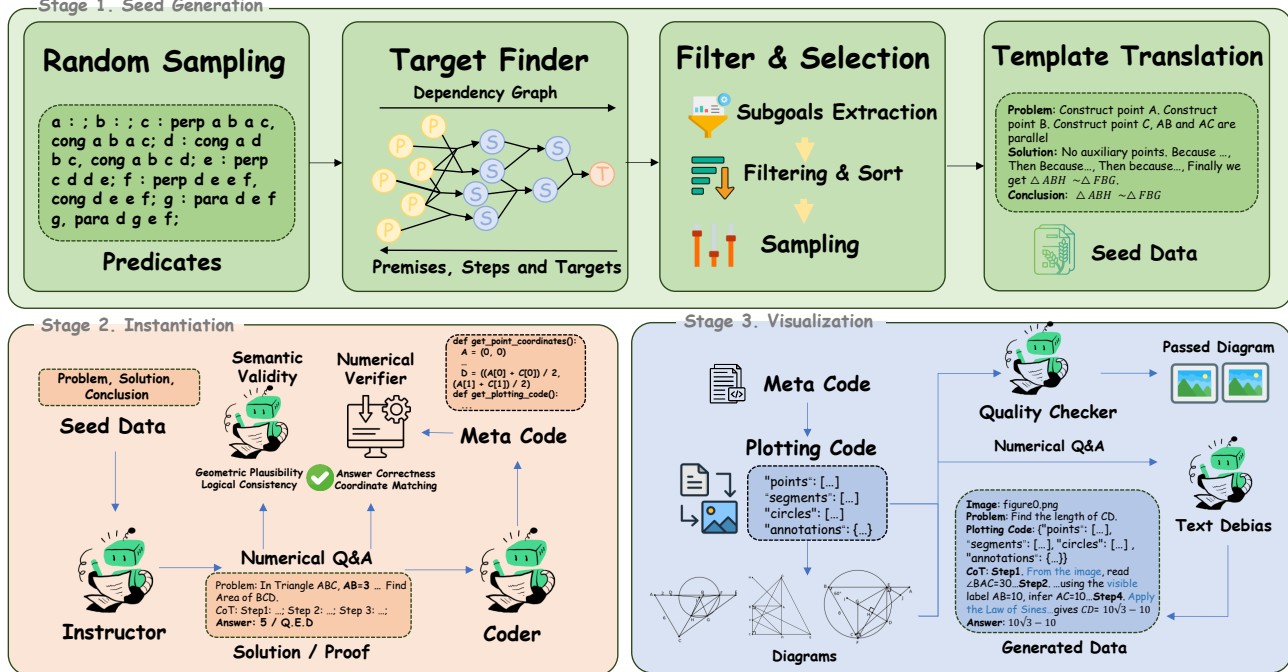

*Figure 2.* **Overview of the data generation pipeline.** Our framework factorizes synthesis into three verifiable stages: (1) Seed Generation for symbolic relational structures, (2) Instantiation for numerical grounding and meta-code generation, and (3) Visualization for diagram rendering and textual debiasing.

**Filter and Selection.** Due to the stochastic nature of predicate sampling, many derived configurations are trivial, degenerate, or pedagogically uninformative. We first discard cases that reduce to tautological or equivalent propositions. For the remaining candidates, we characterize symbolic complexity using two complementary measures: the number of premises, reflecting structural breadth, and the number of proof steps, reflecting deductive depth. We rank all candidates by each measure separately, retain the top $\rho$ percentile under both rankings, and take their intersection as the final seed set. This strategy biases selection toward problems that are simultaneously structurally rich and reasoning-intensive in the symbolic space.

After template-based translation, each retained configuration is represented as *Seed Data* consisting of (i) symbolic premises, (ii) ordered proof steps, and (iii) target conclusions. Notably, these seeds specify only geometric structure and logical relations, and contain no numerical values or coordinate information. They define abstract problem skeletons that will be grounded and verified in the subsequent instantiation stage.

### 3.1.2. INSTANTIATION

This stage maps abstract symbolic seeds into fully specified geometry problems with numerical grounding, natural language formulation, geometric constructions, and verified answers. Concretely, instantiation produces problem statements, reasoning traces, final answers, and geometric construction programs that together define a complete problem instance. To achieve this, we decompose instantiation into three functional components: an Instructor for problem realization, a Coder for geometric construction, and a two-stage verification mechanism for correctness control.

**Instructor for Numerical Grounding and Reasoning Generation.** Given Seed Data consisting of symbolic premises, proof structures, and target conclusions, we employ a reasoning-oriented language model as an *Instructor* to generate instantiated geometry problems. The Instructor assigns concrete numerical values that satisfy all symbolic constraints (e.g., selecting Pythagorean triples for right triangles), translates formal predicates into natural language problem statements, and produces ordered reasoning traces together with final answers. This process transforms abstract relational structures into solvable and pedagogically meaningful geometry questions while preserving all logical dependencies encoded in the seed.

**Coder for Geometric Construction.** For each instantiated problem, we further generate geometric *meta code* using a separate *Coder* model. Conditioned only on the

problem statement, the Coder predicts a set of executable construction functions that compute coordinates of geometric entities through numerical and geometric computations. These functions are executed in a sandbox environment and are parsed to recover two types of information: (i) concrete coordinates of all points, and (ii) structured drawing information, such as which points are connected by segments, which circles are drawn, and what symbolic annotations are present. By executing and parsing these construction functions, we obtain a complete geometric scene specification that supports both numerical verification and deterministic visualization. Importantly, the Coder does not receive the reasoning trace or the final answer as input, preventing leakage of solution information into the geometric construction process. This ensures that all geometric realizations are grounded solely in the textual problem description.

**Two-stage Verification.** To ensure correctness and consistency across modalities, we perform verification at both semantic and geometric levels. **(1) Semantic validation.** An independent language model evaluates whether the problem statement is logically coherent, whether the generated reasoning trace follows from the premises, and whether the stated answer is consistent with the reasoning. **(2) Geometric validation.** Using coordinates produced by executing the meta code, we numerically verify all declared constraints, including lengths, angles, parallelism, and perpendicularity, as well as the final answer when it involves metric quantities. Samples that violate any semantic or geometric condition are discarded.

### 3.1.3. VISUALIZATION

This stage transforms geometric constructions into visual diagrams and removes textual shortcuts that may undermine genuine multimodal reasoning. It consists of two components: diagram rendering based on plotting code, and textual debiasing that enforces reliance on visual information.

**Diagram Rendering.** By executing and parsing the meta code, we obtain structured *plotting code*, including point coordinates and drawing specifications such as segments, circles, and annotations. Using these two types of information, we render geometry diagrams with OpenCV, following standard textbook-style conventions to produce clean and unambiguous figures. In practice, we observe that certain constructions may lead to severe visual overlap or near-degenerate layouts that hinder perception. We therefore perform image-level quality checks and discard samples with excessive overlaps or ambiguous configurations, ensuring that retained diagrams are visually interpretable and suitable for learning.

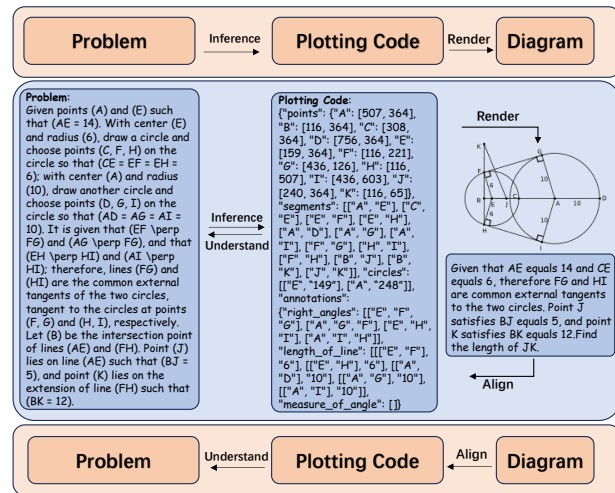

*Figure 3.* **Plotting code as explicit alignment.** Instead of relying on lossy linguistic descriptions, we utilize structured plotting code as an intermediate representation to couple visual perception with symbolic reasoning.

**Textual Debiasing.** Previous studies have shown that models can achieve non-trivial performance on geometry tasks using text alone(Zhang et al., 2024a), suggesting that textual descriptions may act as shortcuts that bypass visual reasoning. Such behavior contradicts the goal of multimodal geometry understanding. To mitigate this issue, we explicitly remove from the question all information that can be directly inferred from the diagram, such as relative positions, intersections, and marked equalities. Moreover, we rewrite the reasoning traces to reflect interactions between visual observations and textual premises, rather than purely symbolic deductions. As a result, solving the problem requires extracting geometric relations from the image instead of relying on redundant textual cues.

### 3.2. Plotting Code as Explicit Alignment

Benefiting from our generation pipeline, each problem instance is associated not only with images, questions, and answers, but also with structured plotting code that explicitly describes the underlying geometric scene. This additional supervision enables us to move beyond answer-level learning and directly supervise geometric perception.

Plotting code serves as a geometry-specific intermediate representation that can be deterministically rendered into diagrams and explicitly encodes geometric entities, constructions, and visual relations. From this perspective, understanding a geometry problem naturally corresponds to recovering its plotting code from the input image.

Based on this observation, we train models to explicitly predict plotting code in addition to reasoning traces and

final answers. Given an input diagram and question, the model is supervised to first reconstruct the geometric scene in the form of plotting code, and then perform reasoning and answer prediction based on the recovered structure. This design makes visual understanding an explicit learning process rather than an implicit latent process.

Compared to answer-only or text-only supervision, plotting-code-based alignment enforces stronger coupling between visual perception and symbolic reasoning, which we empirically show to be critical for robust multimodal geometry reasoning.

# 4. Experiments

In this section, we systematically evaluate the quality of the generated geometry problems and the effectiveness of the proposed alignment strategy. We first describe the implementation and experimental settings in Section 4.1. We then analyze the structural complexity and difficulty of our GeoCode in Section 4.2. Next, we evaluate how well models trained on our synthetic data transfer to public geometry benchmarks in Section 4.3. Finally, we conduct ablation studies to examine the roles of different pipeline components and plotting-code alignment in Sections 4.4 and 4.5.

## 4.1. Implementation Details

**Pipeline Implementation.** We use GenesisGeo (Zhu et al., 2025) for symbolic seed generation and structural selection. Instantiation, reasoning trace generation, semantic validation, textual debiasing, and plotting-code generation are all performed by GPT-OSS-120B (OpenAI, 2025). Diagrams are rendered using OpenCV, followed by image-based quality filtering with Qwen3-VL-32B (Bai et al., 2025a) to remove visually ambiguous samples.

**Training and Evaluation.** All models are trained using LLaMA-Factory (Zheng et al., 2024) with Qwen3-VL-8B-Instruct and Qwen2.5-VL-7B-Instruct as backbones, using LoRA-based SFT for 2 epochs (learning rate $1 \times 10^{-5}$, warmup ratio 0.1). For reinforcement learning, we adopt standard GRPO optimization via the Verl framework (Sheng et al., 2025; Shao et al., 2024), with learning rate $5 \times 10^{-6}$, clip ratio $\epsilon = 0.2$, KL coefficient $\beta = 0.01$, and 4 samples per prompt. All evaluations use identical prompts, greedy decoding ($\tau = 0$), exact-match normalization, and images resized to $224 \times 224$.

## 4.2. Quality and Difficulty of GeoCode

**Data.** GeoCode contains a total of 18k geometry problems, including 13k numerical solution problems and 5k proof-based problems. For numerical problems, we randomly sample 2% of the data to form a held-out test-mini

*Table 1.* **Cross-dataset structural complexity comparison.** Segment counts are extracted under the same protocol and human-validated on 50 samples per dataset. GeoCode shows about 2–3× denser diagram structure than standard benchmarks.

| Dataset | Segments (VLM) | Segments (Human-50) |
|---|---|---|
| Geometry3K | 5.02 | 5.40 |
| GeoQA | 5.42 | 5.04 |
| MathVerse | 4.51 | 4.24 |
| MathVista | 5.16 | 5.44 |
| OlympiadBench | 6.62 | 6.14 |
| GeoCode (ours) | **14.41** | **13.98** |

*Table 2.* **Difficulty comparison with public geometry benchmarks.** Under the same evaluation protocol, Test-mini yields the lowest accuracy for all models. Qwen3-VL denotes Qwen3-VL-32B-Thinking.

| Benchmark | Gemini-2.5-Pro | GPT-5 | Qwen3-VL |
|---|---|---|---|
| Geometry3K | 77.04 | 79.03 | 68.89 |
| GeoQA | 92.04 | 91.91 | 86.60 |
| OlympiadBench | 75.22 | 75.50 | 54.46 |
| Test-mini | **40.67** | **42.16** | **31.72** |

set, which is used for controlled difficulty analysis and comparison with standard benchmarks. All other experiments use disjoint training and test splits.

**Correctness.** All generated samples are filtered by both semantic validation and coordinate geometric verification, ensuring consistency among symbolic relations, numerical constructions, and final answers. To further assess potential residual errors beyond automatic checks, We conduct manual inspection on a random 1% subset of the generated data, and observe no incorrect or inconsistent cases.

**Diversity.** We analyze symbolic seeds to assess structural diversity. Overall, problems exhibit diverse combinations of geometric relations rather than single isolated constraints, and predicate usage follows a long-tailed distribution where frequent relations (e.g., perpendicularity, parallelism) coexist with less frequent but structurally important ones (e.g., cyclic and similarity constructions). This indicates that the dataset is not composed of repetitive templates but covers a wide range of geometric configurations.

**Complexity.** We analyze problem complexity from both structural and difficulty perspectives. **(1) Structure** We perform a unified cross-dataset analysis to compare structural complexity with existing benchmarks. Since prior datasets do not provide explicit structural annotations, we extract geometric structures with Qwen3.5-397B followed by human validation on 50 samples per dataset; for fairness, we apply the same protocol to GeoCode. We use the number of line segments as a representation-invariant proxy for diagram

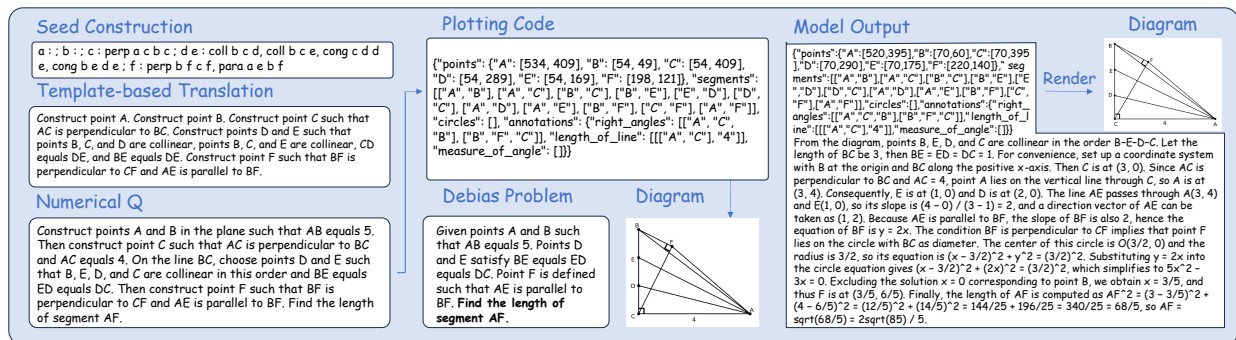

*Figure 4.* **Exemplary data generation and model inference.** The left panel illustrates the end-to-end synthesis process: from Symbolic Seed to Template-based Translation, followed by the Numerical Question, its corresponding Plotting Code, the Debiased Problem statement, and the finally rendered Diagram. The right panel showcases a real-world Inference Example, demonstrating the model's ability to perform structured reasoning and code-based grounding during testing.

| Model | Setting | Geometry3K | MathVerse | MathVista | GeoQA | OlympiadBench | Test-mini |
|---|---|---|---|---|---|---|---|
| Qwen3-VL-8B-Instruct | Baseline | 58.57 | 63.33 | 83.19 | 83.16 | 45.82 | 17.91 |
| | Ours | 60.07 | 65.10 | 84.45 | 84.21 | **57.34** | **26.49** |
| | Δ | +1.50 | +1.77 | +1.26 | +1.05 | **+11.52** | **+8.58** |
| Qwen2.5-VL-7B-Instruct | Baseline | 33.90 | 37.64 | 60.92 | 62.60 | 5.76 | 4.48 |
| | Ours | **40.09** | **46.08** | 57.56 | **66.44** | **6.63** | **13.06** |
| | Δ | **+6.19** | **+8.44** | -3.36 | +3.84 | +0.87 | **+8.58** |

*Table 3.* **Transfer to public geometry and multimodal math benchmarks (%).** We fine-tune Qwen3-VL-8B-Instruct and Qwen2.5-VL-7B-Instruct on GeoCode and report accuracy on Geometry3K, MathVerse, MathVista, GeoQA, OlympiadBench, and Test-mini. *Baseline* and *Ours* denote the instruct checkpoints and the GeoCode-trained models, respectively; Δ is the absolute gain of *Ours* over *Baseline*.

| Model | Base | +SFT | +GRPO |
|---|---|---|---|
| Qwen3-VL | 48 (17.91) | 71 (26.49) | **101 (37.68)** |
| Qwen2.5-VL | 12 (4.48) | 35 (13.06) | **55 (20.52)** |

*Table 4.* **Effect of GeoCode training on Test-mini.** Each cell reports correct answers with accuracy in parentheses; both back-bones improve after Supervised Fine-Tuning (SFT) and further Group Relative Policy Optimization (GRPO).

| Stage | Rejected Samples |
|---|---|
| Semantic validation | 10,510 |
| Geometric verification | 4,171 |
| Plotting execution | 2,017 |
| Image quality filter | 423 |
| Total rejected | 17,121 |
| Final retained | 18,176 |

*Table 5.* **Filtering statistics of the validation pipeline.** The pipeline removes invalid or low-quality samples across semantic, geometric, plotting, and visual-quality checks, retaining 18,176 final examples.

complexity. As shown in Table 1, standard benchmarks typically contain about 4–6 segments per problem, whereas GeoCode reaches 14.41 segments under automatic extraction and 13.98 under human validation. This corresponds to a 2–3× increase in structural density, and the trend is consistent across both automatic extraction and human validation. **(2) Difficulty** To assess problem difficulty and prove complexity is meaningful, we compare model performance on the test-mini set with several standard geometry benchmarks under identical evaluation protocols. As shown in Table 2, test-mini exhibits higher difficulty, reflecting the non-trivial reasoning complexity of the generated problems. Figure 4 qualitatively demonstrates our dataset's complexity and consistency, showcasing the full transition from seed to debiased problem, plotting code, and rendered diagram. More statistics can be found in Appendix E.

### 4.3. Effectiveness on Test-mini and Public Benchmarks

**Evaluation on Test-mini.** We first evaluate models trained on GeoCode on Test-mini, a held-out subset constructed from unseen symbolic seeds and instantiations, which exhibits higher structural complexity and denser geometric constraints than standard benchmarks. To more directly assess the effectiveness of the generated data, we adopt a two-stage training scheme with SFT followed by GRPO. Notably, during the GRPO stage, we apply rewards only on answer correctness and output format, without supervising intermediate reasoning traces or plotting code, so that improvements can be attributed to data quality rather than additional structured supervision. As shown in Ta-

ble 4, both backbones benefit substantially from SFT and further improve with GRPO. For Qwen3-VL-Instruct, accuracy increases from 17.9% to 26.5% after SFT and further to 37.7% after GRPO, while Qwen2.5-VL-Instruct improves from 4.5% to 13.1% and then to 20.5%, respectively. These consistent gains indicate that GeoCode provides strong and stable learning signals even when reinforcement learning is driven solely by final-answer supervision, demonstrating the intrinsic effectiveness of the generated problems.

**Transfer to Public Geometry Benchmarks.** To evaluate out-of-distribution generalization, we further assess models on multiple public benchmarks, including Geometry3K (Lu et al., 2021), MathVerse (Zhang et al., 2024a), MathVista (Lu et al., 2023), GeoQA (Chen et al., 2021), and OlympiadBench (He et al., 2024). For general vision–math benchmarks, we report results only on plane-geometry subsets. As shown in Table 3, models trained on GeoCode outperform their baselines on most benchmarks. In particular, Qwen3-VL-Instruct achieves a large improvement on OlympiadBench (+11.5), one of the most challenging geometry benchmarks, suggesting enhanced robustness on complex multi-step reasoning. Qwen2.5-VL-Instruct also shows notable gains on Geometry3K (+6.2), MathVerse (+8.4), and GeoQA (+3.8). At the same time, the transfer results also reveal a remaining distribution gap. For the weaker Qwen2.5-VL backbone, training on GeoCode slightly hurts MathVista (-3.4) and yields only a modest gain on OlympiadBench (+0.9), indicating that some public benchmarks contain visual styles, question formats, or external reasoning patterns not fully covered by our synthetic distribution. The stronger Qwen3-VL backbone largely mitigates this issue: it remains positive on MathVista (+1.3) and obtains a much larger improvement on OlympiadBench (+11.5), suggesting that stronger base visual perception and reasoning priors better absorb the synthetic-to-real shift and convert GeoCode supervision into transferable capability. Overall, these results indicate that models trained on GeoCode learn transferable geometric representations and reasoning patterns, while also showing that benchmark-specific gaps remain and are more effectively bridged by stronger base models.

### 4.4. Ablation Study on Pipeline Components

**Filtering at Different Stages of the Pipeline.** We track candidate counts at each stage to analyze the contribution of different modules. From 264,705 randomly sampled symbolic structures, only 35,297 remain after seed selection, and 18,176 survive all subsequent stages (Table 5). This reduction mainly reflects the removal of structurally trivial seeds that are unlikely to yield meaningful geometric relations or multi-step reasoning, rather than invalid constructions. Seed selection is computationally lightweight and effectively prunes low-value structures early, without

| Alignment Setting | Test-mini (%) | Geometry3K (%) |
|---|---|---|
| A. None | 17.91 | 58.57 |
| B. Caption | 19.78 | 58.74 |
| C. Caption + Debias | 20.52 | 59.07 |
| D. Code | 22.01 | 59.57 |
| E. Code + Debias | **26.49** | **60.07** |

*Table 6.* **Ablation of alignment supervision strategies.** Code-based supervision outperforms caption-based alignment on both Test-mini and Geometry3K, especially when combined with textual debiasing.

becoming a pipeline bottleneck.

**Verification and Visualization as Quality Gates.** On the remaining candidates, nearly half are further filtered by subsequent stages due to different failure types. Specifically, 10,510 are rejected by semantic validation for inconsistencies among statements, reasoning traces, and answers; 4,171 fail coordinate geometric checks; 2,017 are discarded due to invalid plotting code execution; and 423 are removed by visual quality filtering. As a result, only 18,176 out of 35,297 (51.5%) pass all stages. These failures correspond to distinct error modes spanning linguistic coherence, geometric validity, construction executability, and visual clarity, indicating that the pipeline functions as a sequence of complementary quality gates rather than redundant checks.

### 4.5. Plotting Code Enables Stronger Visual Alignment

To compare different alignment supervision forms, we consider two prediction targets during SFT, plotting code and natural-language captions of the diagram. Captions are generated by Qwen3-VL-32B conditioned on both the rendered diagram and its plotting code. All other inputs are kept identical, and models are trained to regress either code or captions. Results are reported in Table 6. Figure 4 further shows that predicting plotting code before reasoning enables explicit reconstruction of geometric scenes, grounding visual perception in verifiable geometric relations.

**Effectiveness of Code Alignment.** Alignment supervision improves performance over the no-alignment baseline, but different supervision forms behave very differently. Caption supervision brings only small gains on both Test-mini and Geometry3K, even though captions are generated from the same rendered diagrams and plotting code. This suggests that natural-language descriptions provide an incomplete alignment target: they can mention salient objects and relations, but they do not force the model to preserve the full incidence structure, auxiliary constructions, or annotation-level constraints needed for geometric deduction. In contrast, plotting code yields the most substantial gains on both benchmarks because it exposes the diagram as an executable

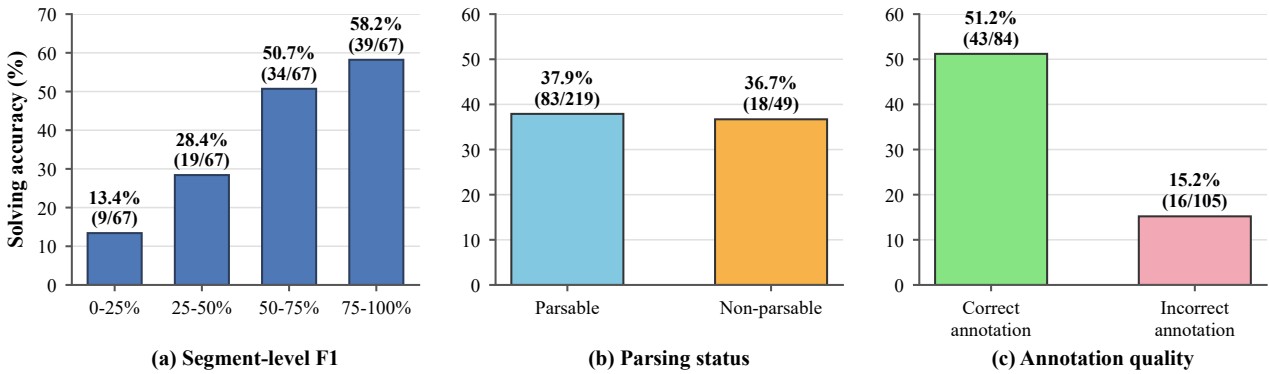

*Figure 5.* **Diagnostic analysis of plotting code as an alignment signal.** (a) Solving accuracy increases monotonically with segment-level F1 between predicted and ground-truth plotting code. (b) Successful parsing of plotting code alone has little correlation with final accuracy. (c) Correct recovery of structural annotations is strongly associated with solving performance.

structural object. Predicting code requires the model to identify points, connect segments, place circles, recover annotations, and preserve the relations among these elements in a form that can be parsed and rendered. We further observe that textual debiasing brings additional gains mainly when combined with code-based supervision, suggesting that debiasing is most useful when the model is already required to ground the image through explicit structural prediction rather than only through free-form text.

**Structural Recovery and Solving Performance.** To verify that the gains from code alignment arise from improved geometric perception rather than only better answer generation, we directly evaluate the structural quality of predicted plotting code on Test-mini, where ground-truth structure is available. We measure segment-level F1 between predicted and ground-truth segments and group samples into equal-sized bins. As shown in Fig. 5(a), solving accuracy increases from 13.4% in the lowest-F1 bin to 58.2% in the highest-F1 bin, indicating that improved recovery of geometric configurations is closely associated with better reasoning performance. The trend is nearly monotonic: as the recovered segment graph becomes closer to the ground-truth construction, the final answer accuracy rises accordingly. This metric is structural rather than outcome-driven: it evaluates whether the predicted code recovers the correct diagram skeleton, independent of how the model verbalizes its reasoning or formats the final answer. Therefore, the diagnostic supports a causal interpretation of the ablation results: code supervision helps because it improves the intermediate visual representation that downstream reasoning depends on.

**Beyond Format Learning.** A natural concern is that plotting code may mainly help because it imposes a rigid output format. However, Fig. 5(b) shows that parsing success is not the bottleneck: parsable outputs achieve 37.9% ac-

curacy, while non-parsable outputs achieve 36.7%. The near-identical performance indicates that syntactic validity alone does not explain the gains from code supervision. In contrast, Fig. 5(c) shows a much sharper separation for structural annotation fidelity. When predicted annotations are fully correct, accuracy reaches 51.2%; when annotations are partially or fully incorrect, accuracy drops to 15.2%. These annotations are not decorative: right-angle marks, equal-length relations, and angle labels often determine which theorem can be applied and which algebraic constraints should be formed. Thus, plotting code is useful not because it teaches a surface format, but because it forces models to recover constraint-bearing visual relations that directly support geometric deduction. This also explains why caption supervision is weaker: a caption may correctly describe the scene at a high level while omitting a small mark or relation that is decisive for the proof, whereas plotting code makes such omissions explicit and learnable.

## 5. Conclusion

We present a pipeline for synthesizing multimodal geometry problems from scratch, with multi-stage validation to ensure consistency among symbolic structures, text, reasoning traces, and diagrams. By decoupling generation into symbolic seed construction, grounded instantiation, and code-based visualization, the pipeline enables scalable synthesis of structurally complex geometry problems. Based on this pipeline, we construct GeoCode, a high-quality dataset pairing each problem with diagrams, solutions, and plotting code, providing explicit supervision for both perception and reasoning. Leveraging the plotting code, we further explore code prediction as an alignment strategy, which encourages models to recover underlying geometric constructions from diagrams and yields more robust visual grounding than text-based supervision, leading to improve performance across multiple geometry benchmarks.

## Acknowledgement

The research is supported by the HKUST startup grant R9895 from CSE, RGC-NSFC project CRS HKUST601/24.

## Impact Statement

This work aims to advance multimodal machine learning by improving visual–symbolic alignment and reasoning in geometry, a core domain for studying structured multimodal understanding. Potential positive impacts include supporting research on multimodal reasoning, enabling better educational tools for geometry learning, and facilitating future studies on interpretable visual–symbolic models.

All data used in this work are synthetically generated through automated symbolic and geometric procedures, and do not involve personal data, sensitive content, or real-world imagery, thereby posing no privacy or data protection concerns.

As with many advances in reasoning-capable language and vision–language models, there is a general risk that improved problem-solving abilities could be misused for academic dishonesty or automated answer generation. However, this risk is not specific to our method and is shared by most progress in large-scale reasoning models. We believe that the primary and intended use of our work is to support research and education, and we do not foresee significant negative societal consequences specific to this contribution.

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

# A. Discussion

In this section, we compare our approach with existing methods and discuss future directions. Our work focuses on synthesizing multimodal geometry data and enabling effective visual alignment. We first examine related data synthesis methods, then highlight our core innovations, and finally outline promising future directions.

**Comparison with Existing Data Synthesis Methods.** Early data synthesis approaches for geometry reasoning primarily focused on augmenting existing datasets rather than generating new visual content. G-LLaVA (Gao et al., 2023) represents one of the earliest data synthesis methods in this domain, but it does not generate new images; instead, it augments datasets through semantic modifications and explores text-based visual geometric alignment using predicate-based methods. Subsequent works have advanced toward generating novel images. R-CoT (Deng et al., 2024) employs a sampling-based approach to select geometric configurations and synthesizes geometry data using large language models. GeoGen (Pan et al., 2025) leverages the symbolic engine FormalGeo (Zhang et al., 2023) to search in symbolic space and translates results using LLMs. TrustGeoGen (Fu et al., 2025) represents a particularly powerful approach that exclusively searches in symbolic space using the AlphaGeometry engine to synthesize data. MathCoder-VL (Wang et al., 2025) introduces an interesting variation by adjusting temperature parameters to generate novel figures through code, synthesizing instruction pairs using LLMs, and validating correctness through A/B testing. We acknowledge the valuable insights these works have provided for our research.

Compared to these approaches, our method introduces several core innovations. *First*, we decompose the synthesis process into three distinct stages, factorizing geometry into basic structure, instance information, and visualization code. This decomposition is not only more intuitive but has also proven more effective in practice, enabling the generation of more diverse structures and leveraging LLM priors to produce interesting problems. From this perspective, our approach differs fundamentally from the aforementioned works. *Second*, code-based visualization provides complete geometric information, allowing us to thoroughly explore the role of visual modalities. Through debiasing and explicit alignment methods, we promote visual understanding, representing a deeper investigation beyond synthesis. Unlike G-LLaVA (Gao et al., 2023), MAVIS (Zhang et al., 2024b), and EAGLE (Li et al., 2024), which employ different alignment strategies, our approach uses plotting code for alignment, which we empirically demonstrate to be more effective. *Third*, we acknowledge that LLM hallucinations pose challenges for synthesizing mathematical problems, necessitating multiple validation mechanisms. Therefore, we implement multi-level verification through semantic, numerical, and visual checks to ensure the correctness of multimodal problems. In particular, our code-based verification represents a significant step forward in synthesizing mathematical geometry problems.

**Future Directions.** Looking ahead, we believe that data synthesis methods will inspire more downstream tasks, particularly in areas such as Thinking With Images, enabling more effective supervision or reinforcement learning algorithm development. However, data synthesis represents only the first step. We also hope that code-based alignment approaches can find broader applications, where explicit structural grounding can bridge the gap between visual perception and symbolic reasoning.

# B. Training and Evaluation Details

This section provides detailed specifications of our training and evaluation procedures, including supervised fine-tuning (SFT) and reinforcement learning with GRPO.

## B.1. Supervised Fine-Tuning

We perform SFT using cross-entropy loss on our synthetic geometry dataset. The training configuration is as follows:

**Model Configuration.** We use Qwen2.5-VL-7B-Instruct and Qwen3-VL-8B-Instruct as base models, both following identical training configurations. The vision tower is frozen during training, while the multi-modal projector and language model are fine-tuned. We employ LoRA with rank 128, applied to all trainable parameters. The maximum image resolution is set to $512 \times 512$ pixels (262,144 total pixels).

**Training Hyperparameters.** The model is trained for 2 epochs with a learning rate of $1.0 \times 10^{-5}$ using cosine learning rate scheduling. We use a warmup ratio of 0.1, batch size of 4 per device with gradient accumulation over 8 steps (effective batch size of 32), and bfloat16 precision. The maximum sequence length is set to 8192 tokens. We train on 18k samples

from our synthetic dataset.

**Loss Function.** During SFT, we minimize the standard cross-entropy loss:

$$\mathcal{L}_{\text{SFT}} = -\sum_{t=1}^{T} \log P(y_t \mid y_{<t}, x, I),$$

where $x$ is the input text prompt, $I$ is the input image, $y_t$ is the token at position $t$, and $T$ is the total sequence length. The model is supervised to predict plotting code, reasoning traces, and final answers.

### B.2. Reinforcement Learning with GRPO

After SFT, we further optimize the model using Group Relative Policy Optimization (GRPO) (Shao et al., 2024), a variant of PPO that groups multiple responses for more stable policy updates.

**GRPO Objective.** The GRPO objective function is defined as:

$$\mathcal{L}_{\text{GRPO}} = \mathbb{E}_{(x,I)\sim\mathcal{D}} \left[ \frac{1}{G} \sum_{g=1}^{G} \sum_{i=1}^{n_g} (r(x, I, y_{g,i}) - \bar{r}_g) \log \pi_\theta(y_{g,i} \mid x, I) - \beta \text{KL}(\pi_\theta \| \pi_{\text{ref}}) \right],$$

where $G$ is the number of groups, $n_g$ is the number of responses in group $g$, $r(x, I, y_{g,i})$ is the reward for response $y_{g,i}$, $\bar{r}_g = \frac{1}{n_g} \sum_{i=1}^{n_g} r(x, I, y_{g,i})$ is the group mean reward, $\pi_\theta$ is the current policy, $\pi_{\text{ref}}$ is the reference policy (the SFT model), and $\beta = 0.01$ is the KL penalty coefficient.

**Reward Function.** We employ a composite reward function that combines format correctness and answer accuracy:

$$r(x, I, y) = r_{\text{format}}(y) + r_{\text{answer}}(x, I, y),$$

where $r_{\text{format}}(y) \in \{0, 1\}$ is a binary reward for correct output format, and $r_{\text{answer}}(x, I, y) \in \{0, 1\}$ is a binary reward for answer correctness.

The format reward $r_{\text{format}}(y)$ requires the model output to follow the structure with three explicitly delimited blocks:

$$\texttt{}\, C \,\texttt{} \,\texttt{<think>}\, R \,\texttt{</think>} \,\texttt{<answer>}\, A \,\texttt{</answer>},$$

where $C$ is the plotting code, $R$ is the reasoning trace (placed inside `<think> </think>`), and $A$ is the final answer (placed inside `<answer> </answer>`). The format reward is 1 if the output contains all three required components in the correct order with correct opening and closing tags, and 0 otherwise.

The answer reward $r_{\text{answer}}(x, I, y)$ is 1 if the extracted answer $A$ matches the ground truth answer (with numerical tolerance for floating-point comparisons), and 0 otherwise.

**Training Configuration.** We train for 1 epoch with a learning rate of $5.0 \times 10^{-6}$. The training batch size is 256, with PPO mini-batch size of 128 and micro-batch size of 8 per GPU. We sample $n = 4$ responses per prompt for reward estimation. The KL loss uses the low-variance KL estimator with coefficient $\beta = 0.01$, and the clip ratio is set to $\epsilon = 0.2$. We use FSDP (Fully Sharded Data Parallel) for distributed training across 4 H200 GPUs, with gradient checkpointing enabled to reduce memory usage.

**Implementation Details.** The rollout and reference models use tensor model parallelism with size 2. The rollout engine uses vLLM with GPU memory utilization set to 0.6. We disable the multi-modal preprocessor cache and use eager execution mode. The maximum prompt and response lengths are both set to 2048 tokens.

### B.3. Evaluation Protocol

**Test Set Construction.** For non-geometry-specific datasets, we follow the original dataset annotations and select subsets that involve plane geometry problems. Table 7 summarizes the sizes of these geometry-focused minisets extracted from each benchmark.

| Dataset | Test Set Size |
|---|---|
| Geometry3K | 601 |
| GeoQA | 754 |
| OlympiadBench | 347 |
| MathVerse | 510 |
| MathVista | 238 |

*Table 7.* **Evaluation subset sizes.** For general multimodal math datasets, we keep only plane-geometry problems.

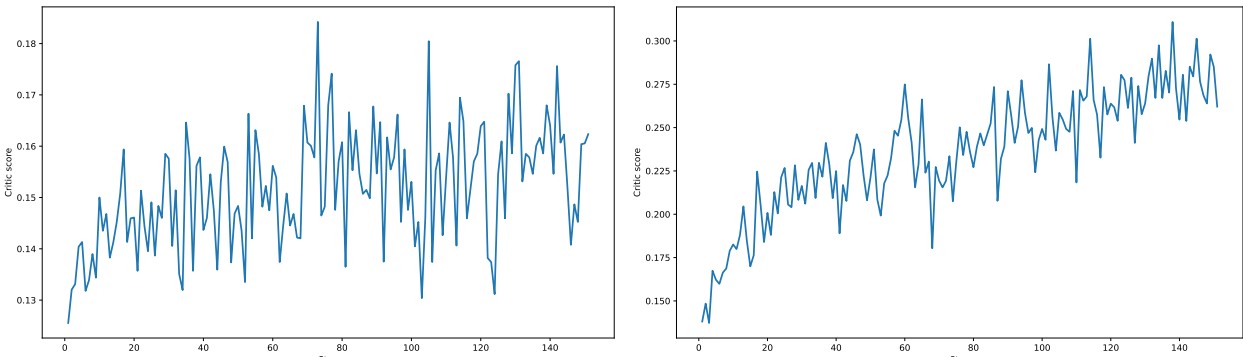

*Figure 6.* **GRPO reward curves for different backbones.** Qwen2.5-VL (Bai et al., 2025b) (left) exhibits slower reward growth with larger fluctuations, while Qwen3-VL (Bai et al., 2025a) (right) shows a more stable and monotonic improvement, indicating stronger geometric reasoning ability and better alignment with our hard, structurally complex data.

**Evaluation Details.**   To ensure fair and consistent evaluation across all models, we adopt a standardized evaluation protocol. We set the temperature parameter $\tau = 0$ for all evaluations, enabling deterministic greedy decoding. All models use the same system prompt: `You are a mathematical reasoning assistant.` `Your task is to solve the problem and give the correct answer.`

For answer extraction and matching, we parse answers from either `<answer>` or `<box>` tags in the model output. We then process the extracted answers through LaTeX parsing and symbolic evaluation using SymPy to compute numerical values. An answer is considered correct if the computed value matches the ground truth answer within a tolerance of $10^{-6}$. This numerical matching approach ensures robust evaluation across different answer formats and representations.

To maintain fairness, all models are evaluated using identical evaluation procedures, including the same answer extraction logic, parsing methods, and matching criteria. We evaluate only problems that require solution generation (i.e., solution-type problems), excluding other problem types from the evaluation.

## C. Effect of Backbone Models

By constructing hard and structurally complex geometry instances that require multi-step reasoning and sophisticated spatial understanding, our method provides a natural testbed for evaluating how different vision-language backbones respond to reinforcement learning optimization. We compare Qwen2.5-VL (Bai et al., 2025b) and Qwen3-VL (Bai et al., 2025a) under identical GRPO training configurations, examining both their final performance on Test-mini (Table 4) and their training dynamics through reward curves (Figure 6).

The results reveal a clear distinction between the two backbones. Qwen3-VL (Bai et al., 2025a), which has been extensively pre-trained on STEM and geometry-related corpora, demonstrates superior geometric reasoning capabilities: during GRPO training, its average reward exhibits steady, monotonic growth and converges to a high, stable plateau (Figure 6, right). This behavior suggests that Qwen3-VL can effectively leverage the structural complexity and difficulty of our generated data to drive consistent policy improvement. In contrast, while Qwen2.5-VL (Bai et al., 2025b) also benefits from GRPO and achieves notable gains (from 13.1% to 20.5% on Test-mini), its reward curve displays slower growth and significantly larger variance throughout training (Figure 6, left), indicating less stable optimization and lower sample efficiency.

This comparison yields two important insights. First, backbone selection is critical for vertical domains like geometry, where

| Model | Baseline | OOD | Ours | Δ (Ours–OOD) |
|---|---|---|---|---|
| Qwen3-VL-Instruct | 83.16 | 84.21 | **89.66** | +5.45 |
| Qwen2.5-VL-Instruct | 62.60 | 66.44 | **68.83** | +2.39 |

*Table 8.* **Text-seeded synthesis on GeoQA.** "Baseline" uses pretrained backbones, "OOD" trains only on GeoCode, and "Ours" adds GeoQA-style text-seeded variants, improving both backbones over OOD training.

specialized pre-training on domain-relevant data substantially facilitates downstream reinforcement learning. Second, the high difficulty and structural richness of our automatically generated data serve as an effective probe that can distinguish between strong backbones, validating both the quality of our data generation pipeline and its utility for benchmarking geometric reasoning capabilities.

## D. Extensibility of Our Pipeline

Our generation pipeline is divided into three stages: symbolic seed generation, instance instantiation, and visualization. The first stage lays down the underlying geometric structure, while the latter two stages materialize concrete problem instances and diagrams. This modular design naturally raises the question of whether the pipeline can be extended to *ingest* existing textual problems and synthesize new data that reduces the domain gap between our synthetic distribution and established benchmarks.

To explore this direction, we examine common geometry datasets in the community. Geometry3K provides relatively sparse textual descriptions, which often lack sufficient structural information to serve as reliable seeds for our symbolic generator. In contrast, GeoQA contains rich and highly structured textual statements that implicitly encode geometric configurations and reasoning chains, for example: "In quadrilateral $ABCD$, $\angle B$ is 40° larger than $\angle A$; what is the measure of $\angle D$?" or "In $\triangle ABC$, points $D$ and $E$ lie on $AB$ and $AC$ with $DE \parallel BC$ and $\frac{AD}{AB} = \frac{3}{7}$; given $AE = 6$, what is the length of $EC$?" These formulations clearly expose the roles of key entities and constraints, making them suitable as structural backbones.

Motivated by this observation, we propose an extended pipeline where we reuse the textual stems of existing training problems as high-level structural seeds, and then apply our instantiation and visualization stages to generate variants with concrete coordinates and diagrams. This "text-to-structure" augmentation bridges part of the domain gap by aligning the combinatorial structure of our synthetic data with that of real benchmarks, while still benefiting from automatic validation and visualization.

We evaluate this approach in an out-of-distribution (OOD) setting on GeoQA, comparing three training regimes: (i) *Baseline*, using only the original pretrained backbones; (ii) *OOD*, training on our original synthetic dataset; and (iii) *Ours*, further augmenting training with text-seeded synthetic data derived from GeoQA-style problems. Results are summarized in Table 8.

Overall, these results demonstrate that our solver-driven pipeline is not limited to fully synthetic symbolic seeds, but can flexibly absorb real-world textual problems as structural inputs. This extensibility enables principled adaptation to diverse geometry datasets and provides a general recipe for combining symbolic generation with existing benchmarks.

## E. Example Data from Our Pipeline

This section presents example diagrams generated by our pipeline to illustrate the types of geometric structures it produces.

Figure 7 provides statistical insights into the predicate distribution of our generated problems. The left panel shows the frequency distribution of predicates sampled during seed generation, revealing a long-tail distribution where commonly used predicates such as perpendicularity and parallelism occur with much higher probability than less frequent ones like concyclic or similarity relations. This distribution pattern demonstrates that our generation process is not a rigid template-based procedure, but rather a flexible process that naturally captures the varying frequencies of different geometric relationships in real-world problems. The right panel displays the distribution of predicate counts per problem, which follows an approximately normal distribution centered around a mean of 10 predicates. This level of structural complexity, with an average of 10 geometric constraints per problem, significantly exceeds the complexity typically found in existing geometry datasets, as can be observed in the detailed examples shown in Figure 8.

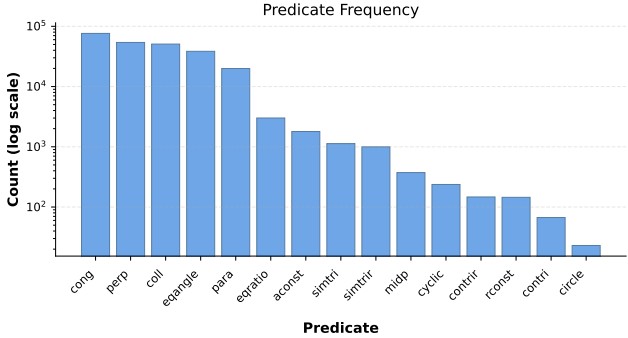
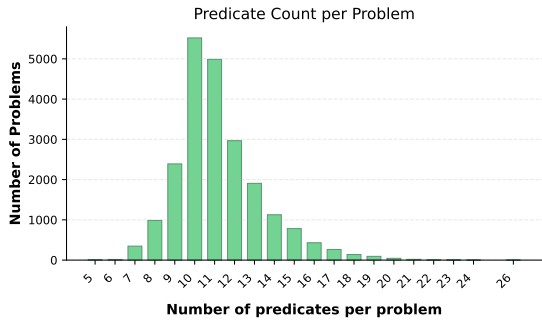

*Figure 7.* Predicate statistics of generated geometry problems: predicate frequency (left) and predicate count per problem (right).

Figure 8 shows 20 example diagrams from our fully synthetic dataset, while Figure 9 displays 20 example diagrams generated using the extended approach described in Appendix D(text-seeded synthesis).

The diagrams in Figure 8 demonstrate the complex geometric structures that our fully synthetic pipeline can generate. These examples showcase several key characteristics of our solver-driven generation approach. First, the diagrams exhibit **dense geometric constructions** with multiple overlapping circles, intricate polygon configurations, and complex intersection patterns that require sophisticated spatial reasoning to fully comprehend. Second, they feature **non-trivial relational constraints** including rich networks of geometric relations such as parallelism, perpendicularity, cyclic quadrilaterals, similarity constructions, and congruence relations that create multi-step reasoning chains. Third, the problems involve **high entity density** with numerous points, segments, and circles that have interwoven dependencies, challenging both visual perception and symbolic reasoning capabilities. The structural richness of these diagrams reflects the effectiveness of our pipeline in exploring complex geometric configurations through symbolic seed generation, dependency construction, and rigorous verification mechanisms.

The diagrams in Figure 9 illustrate the output of our extended pipeline when applied to existing textual problem formulations. These examples demonstrate the compatibility and extensibility of our framework, showing that the instantiation and visualization stages can effectively process different types of input sources. By feeding textual problem statements into our pipeline, we can leverage the same verification mechanisms and visualization capabilities to produce geometrically consistent diagrams, even when the underlying structural seeds originate from existing problem formulations rather than purely symbolic generation. This flexibility highlights the modular design of our pipeline, where the instantiation and visualization components can operate independently of the seed generation stage, enabling adaptation to diverse data sources while maintaining strict geometric correctness and visual quality standards.

Together, these example diagrams illustrate the range of geometric structures that our pipeline can produce, from complex solver-driven constructions to text-grounded instantiations, demonstrating both the effectiveness of our fully synthetic approach and the compatibility of our framework with alternative input modalities.

## F. Detailed Pipeline and Algorithms

Our complete generation pipeline is summarized in Algorithm 1. In this section, we provide detailed descriptions of the coordinate verification mechanism, which ensures geometric consistency between symbolic constraints, numerical coordinates, and visual diagrams.

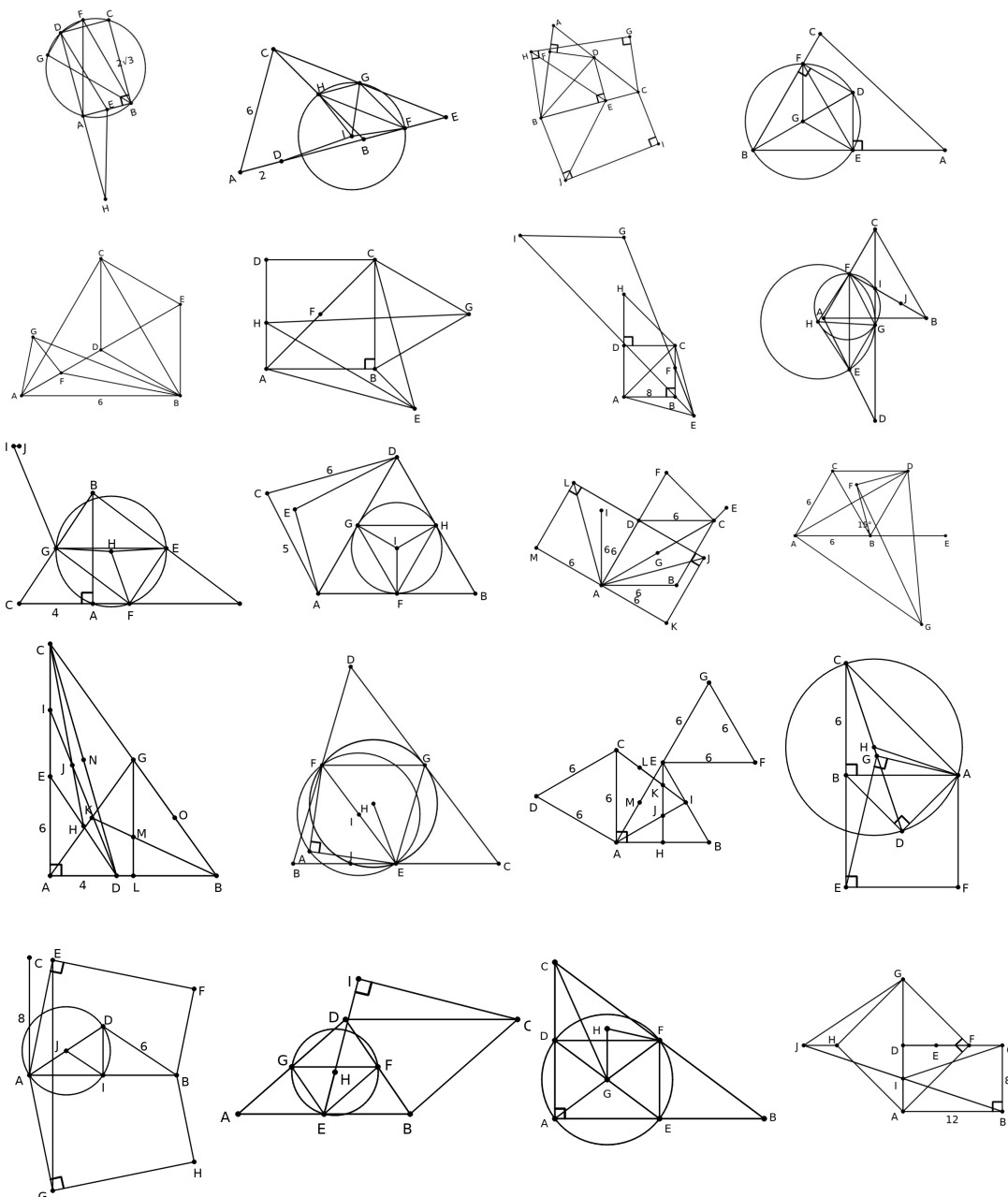

*Figure 8.* **Example diagrams from our fully synthetic pipeline.** These 20 examples illustrate the geometric structures generated by our solver-driven pipeline, featuring dense relational constraints, multi-step constructions, and non-trivial geometric configurations. Each diagram exhibits rich structural complexity with multiple geometric entities, intricate spatial relationships, and sophisticated reasoning patterns.

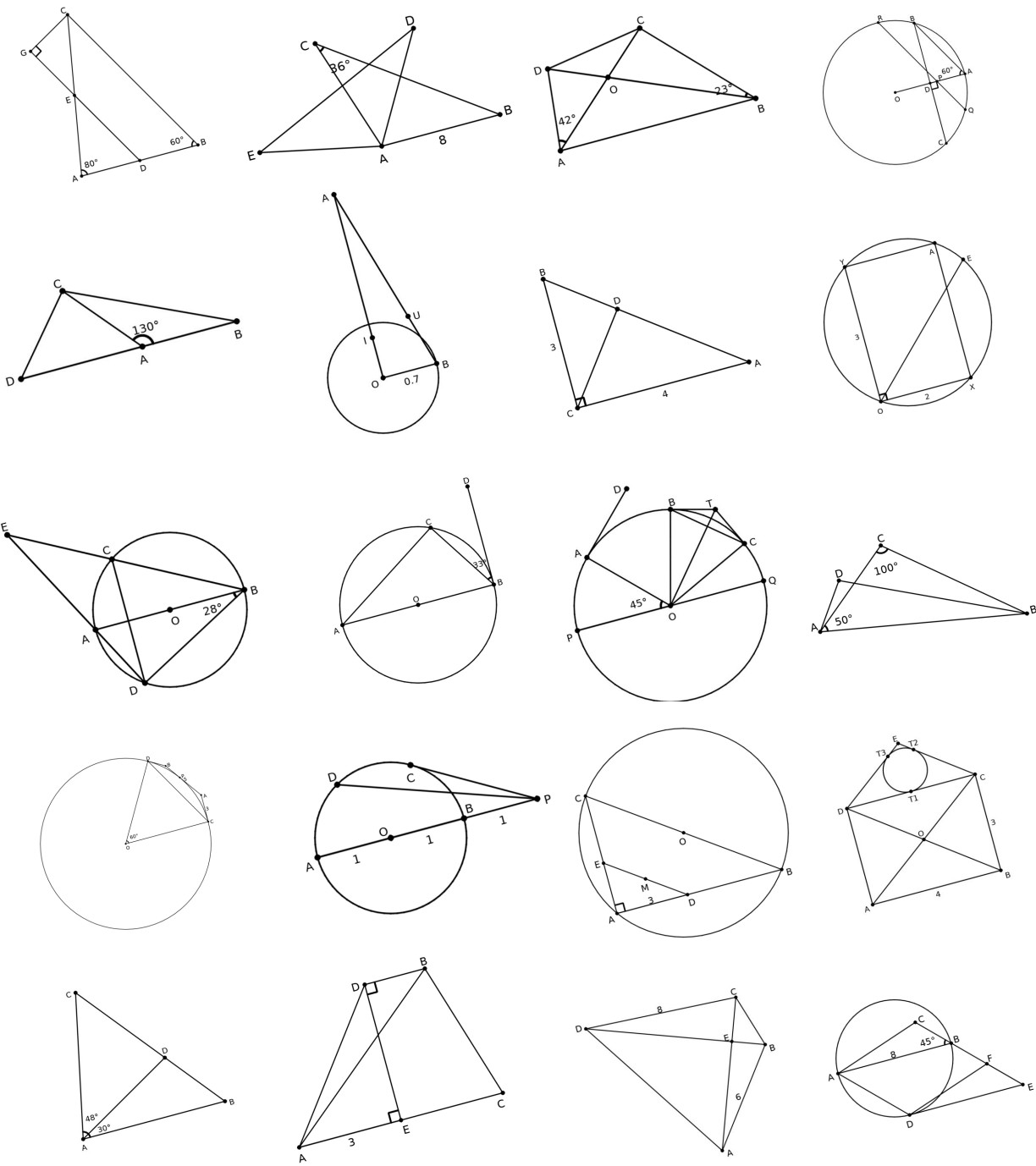

*Figure 9.* **Example diagrams from text-seeded synthesis (Appendix E).** These 20 examples illustrate the geometric structures generated by feeding existing textual problem statements into our instantiation and visualization stages.

---

**Algorithm 1** End-to-end pipeline for synthesizing multimodal geometry problems from scratch

---

**Input:** Predicate pool $\mathcal{P}$; theorem library $\mathcal{T}$;
number of sampled predicate sets $N$; target/subgoal budget $M$;
selection rule $\mathsf{SelectSeed}(\cdot)$;
Instructor LLM $\mathcal{L}_{\text{inst}}$; Coder LLM $\mathcal{L}_{\text{code}}$;
Semantic checker LLM $\mathcal{L}_{\text{sem}}$; quality checker $\mathsf{ImgQC}(\cdot)$.
**Output:** Final dataset $\mathcal{D} = \{(I, P', R', A, C_{\text{plot}})\}$, where $I$ is diagram, $P'$ is debiased problem, $R'$ is debiased CoT, $A$ is answer, and $C_{\text{plot}}$ is plotting code.

$\mathcal{D} \leftarrow \emptyset$;                                                  `// final retained samples`
**for** $i = 1$ **to** $N$ **do**

    `// Stage 1: Seed generation from symbolic predicates`
    $S \leftarrow \mathsf{SamplePredicates}(\mathcal{P})$;                      `// randomly sample a predicate set`
    $G \leftarrow \mathsf{BuildDependencyGraph}(S; \mathcal{T})$;                        `// dependency graph`
    $\mathcal{G} \leftarrow \mathsf{ExtractSubgoals}(G)$;                  `// candidate targets/subgoals from proofs`
    $\mathcal{G} \leftarrow \mathsf{FilterTrivial}(\mathcal{G})$;                `// remove tautologies/degeneracies`
    $\mathcal{G}^\star \leftarrow \mathsf{SampleSubgoals}(\mathcal{G}, M)$;              `// subgoal sampling/budgeting`
    $\mathcal{S}_{\text{seed}} \leftarrow \mathsf{SelectSeed}(S, G, \mathcal{G}^\star)$;                `// premises, steps, targets`
    **if** $\mathcal{S}_{\text{seed}} = \emptyset$ **then**
        $\llcorner$ **continue**

    `// Stage 2: Instantiation by Instructor + semantic validation`
    $(P, R, A) \leftarrow \mathcal{L}_{\text{inst}}(\mathcal{S}_{\text{seed}})$;                            `// problem, CoT, answer`
    $ok_{\text{sem}} \leftarrow \mathcal{L}_{\text{sem}}(P, R, A)$;                  `// logical/format/consistency check`
    **if** $ok_{\text{sem}} = \textit{false}$ **then**
        $\llcorner$ **continue**

    `// Stage 2 (cont.): Meta code generation + executable grounding`
    $C_{\text{meta}} \leftarrow \mathcal{L}_{\text{code}}(P)$;                  `// meta construction code conditioned only on P`
    $ok_{\text{exec}} \leftarrow \mathsf{ExecuteSandbox}(C_{\text{meta}})$;                      `// safety + runtime validity`
    **if** $ok_{\text{exec}} = \textit{false}$ **then**
        $\llcorner$ **continue**
    $(\mathbf{X}, \Pi) \leftarrow \mathsf{ParseMetaCode}(C_{\text{meta}})$;                  `// X: coordinates; Π: plotting code/spec`
    $ok_{\text{geo}} \leftarrow \mathsf{VerifyGeometry}(P, \mathbf{X})$;                  `// verify declared constraints numerically`
    $ok_{\text{ans}} \leftarrow \mathsf{VerifyAnswer}(P, A, \mathbf{X})$;                  `// check metric answer via coordinates`
    **if** $ok_{\text{geo}} = \textit{false}$ **or** $ok_{\text{ans}} = \textit{false}$ **then**
        $\llcorner$ **continue**

    `// Stage 3: Visualization + image quality filtering`
    $C_{\text{plot}} \leftarrow \mathsf{ToPlottingCode}(\Pi)$;          `// serialize plotting code into plotting code DSL/JSON`
    $I \leftarrow \mathsf{Render}(C_{\text{plot}})$;                      `// deterministic diagram rendering`
    $ok_{\text{img}} \leftarrow \mathsf{ImgQC}(I)$;                      `// overlap/ambiguity/clutter checks`
    **if** $ok_{\text{img}} = \textit{false}$ **then**
        $\llcorner$ **continue**

    `// Stage 3 (cont.): Textual debiasing using plotting code`
    $(P', R') \leftarrow \mathsf{Debias}(P, R, \Pi)$;              `// remove diagram-inferable cues; rewrite reasoning trace`
    $\mathcal{D} \leftarrow \mathcal{D} \cup \{(I, P', R', A, C_{\text{plot}})\}$
**return** $\mathcal{D}$.

---

### F.1. Coordinate Verification Mechanism

The coordinate verification stage performs numerical validation to ensure that the generated coordinates satisfy all geometric constraints declared in the problem statement. This verification is critical for maintaining consistency across symbolic structure, textual description, and visual representation.

**Coordinate Generation.** Given a problem statement $P$, the Coder LLM $\mathcal{L}_{\text{code}}$ generates Python code that computes coordinates for all geometric entities mentioned in $P$. The generated code must satisfy several constraints: (1) all points must have distinct coordinates to avoid degeneracy, (2) all geometric relations (e.g., perpendicularity, parallelism, collinearity) must be explicitly enforced through numerical construction, and (3) the code must be executable in a sandboxed environment without external dependencies beyond standard mathematical libraries. The code is executed to obtain a coordinate dictionary $\mathbf{X} = \{p_i : (x_i, y_i)\}$, where each $p_i$ is a point label and $(x_i, y_i)$ are its 2D coordinates.

**Geometric Constraint Verification.** After extracting coordinates $\mathbf{X}$ and plotting specifications $\Pi$ from the executed meta code, we perform two types of numerical checks:

**(1) Constraint Verification** (VerifyGeometry): For each geometric relation explicitly stated in the problem $P$, we verify that the coordinates satisfy the corresponding numerical condition. This includes:

- **Length constraints**: For any segment $AB$ with declared length $l$, we check that $|\|\mathbf{X}(A) - \mathbf{X}(B)\| - l| < \epsilon$, where $\epsilon = 10^{-6}$ is a numerical tolerance.

- **Angle constraints**: For any angle $\angle ABC$ with declared measure $\theta$, we compute the angle using vector dot product and verify $|\angle(\mathbf{X}(A), \mathbf{X}(B), \mathbf{X}(C)) - \theta| < \epsilon$.

- **Parallelism and perpendicularity**: For parallel lines $AB \parallel CD$, we verify that the angle between direction vectors is within $\epsilon$ of $0°$ or $180°$; for perpendicular lines $AB \perp CD$, we verify the angle is within $\epsilon$ of $90°$.

- **Collinearity**: For collinear points $A, B, C$, we verify that the area of triangle $ABC$ (computed via cross product) is within $\epsilon$ of zero.

- **Circle constraints**: For a circle with center $O$ and radius $r$, or a circle passing through points $A, B, C$, we verify that all declared points on the circle satisfy $|\|\mathbf{X}(p) - \mathbf{X}(O)\| - r| < \epsilon$ (or equivalent for three-point circles).

**(2) Answer Verification** (VerifyAnswer): For problems with numerical answers, we evaluate the answer quantity using the generated coordinates. Specifically, we parse the DSL expression from $\Pi$.quantities (e.g., `length(A, B)`, `angle(A, B, C)`, or more complex expressions like `length(A, B) − length(C, D)` for proof problems). The DSL supports geometric primitives including:

- Point-based quantities: `length(A, B)`, `angle(A, B, C)`, `area(A, B, C, ...)`, `perimeter(A, B, C, ...)`.

- Line-based quantities: `angle_between_lines(A, B, C, D)` and its trigonometric variants.

- Circle-based quantities: `radius(C1)`, `arc_length(C1, A, B)`, `sector_area(C1, A, B)`, etc., where $C1$ is a circle ID.

Each DSL expression is evaluated by substituting point coordinates and circle parameters, then compared against the declared answer $A$ with tolerance $\epsilon$.

For **computation problems**, the DSL expression directly represents the quantity to be computed (e.g., `length(A, B)` for finding the length of segment $AB$), and we verify that the evaluated value matches the declared answer $A$.

For **proof problems**, we transform the statement to be proved into a zero-value expression for numerical verification. Specifically, if the conclusion to be proved is an equality relation (e.g., "$AB = CD$", "$\angle ABC = \angle DEF$", or "$AB \parallel CD$"), we convert it into a difference expression that should evaluate to zero:

- Equality of lengths: "$AB = CD$" $\rightarrow$ `length(A, B) − length(C, D)`

- Equality of angles: "$\angle ABC = \angle DEF$" $\rightarrow$ `angle(A, B, C) − angle(D, E, F)`

- Parallelism: "$AB \parallel CD$" $\rightarrow$ `angle_between_lines(A, B, C, D) − 0`

- Perpendicularity: "$AB \perp CD$" $\rightarrow$ `angle_between_lines(A, B, C, D) − 90`

- Collinearity: "Points $A, B, C$ are collinear" $\rightarrow$ `area(A, B, C) − 0`

- Ratio relations: "$AB : CD = EF : GH$" $\rightarrow$ `length(A, B)/length(C, D) − length(E, F)/length(G, H)`

We then evaluate the transformed expression using the generated coordinates and verify that the result is within $\epsilon$ of zero, confirming that the geometric relation holds numerically. This zero-value verification approach enables rigorous validation of proof statements through coordinate-based computation, ensuring that the symbolic conclusion is consistent with the numerical instantiation.

**Annotation Consistency Check.** Beyond constraint and answer verification, we also validate that annotations in $\Pi$.`annotations` are consistent with the computed coordinates:

- **Length annotations**: For each entry $[["A","B"], v]$ in `length_of_line`, we verify that the actual length $\|\mathbf{X}(A) - \mathbf{X}(B)\|$ matches the annotated value $v$ (after parsing LaTeX expressions like $2\sqrt{3}$).

- **Right angle annotations**: For each entry $["A","B","C"]$ in `right_angles`, we verify that $\angle(\mathbf{X}(A), \mathbf{X}(B), \mathbf{X}(C)) = 90° \pm \epsilon$.

- **Angle measure annotations**: For each entry $[["A","B","C"], \theta]$ in `measure_of_angle`, we verify that the computed angle matches $\theta$ (with appropriate unit conversion if the annotation uses radians).

Samples that fail any of these verification steps are discarded, ensuring that all retained instances maintain strict numerical consistency between symbolic constraints, coordinate realizations, and visual diagrams. This multi-level validation mechanism is essential for generating high-quality geometry problems with guaranteed mathematical correctness.

### F.2. Plotting Code Format Specification

After coordinate verification, the geometric construction is serialized into a structured *plotting code* format that serves as the interface between coordinate computation and diagram rendering. This format is designed to be both human-readable and machine-executable, enabling deterministic visualization while preserving all geometric information necessary for verification and reasoning.

The plotting code is represented as a JSON-like dictionary structure with the following schema:

**Point Coordinates.** The `points` field maps point labels to 2D coordinates:

```
"points": {
    "A": (xA, yA),
    "B": (xB, yB),
    ...
}
```

Each coordinate pair $(x_i, y_i)$ must be a tuple or list of two numeric values (integers or floats). All points must have distinct coordinates to avoid degeneracy.

**Segments.** The `segments` field lists all line segments to be drawn:

```
"segments": [
    ["A", "B"],
    ["B", "C"],
    ...
]
```

Each segment is represented as a two-element list containing the labels of its endpoints. Segments are drawn as straight lines connecting the specified points.

**Circles.** The `circles` field specifies all circles in the diagram. Each circle is represented as a list with the following formats:

- `["C1", "O", r]`: Circle with ID `C1`, center at point `O`, and radius $r$ (numeric value).

- `["C2", "O", "P"]`: Circle with ID `C2`, center at point `O`, and radius equal to the distance from `O` to `P`.

- `["C3", "A", "B", "diameter"]`: Circle with ID `C3`, where segment `AB` is the diameter.

- `["C4", "A", "B", "C"]`: Circle with ID `C4` passing through three points `A`, `B`, and `C`.

Each circle must have a unique ID (e.g., `C1`, `C2`, `C3`) that is used for referencing in DSL expressions. The circle ID is distinct from point labels and is required for all circle-related geometric quantities.

In the practical implementation of our code-based alignment, we simplify the regression target for circular entities. Although our data generation pipeline supports multiple construction methods for circles (e.g., via diameter or three points), we unify these into a standard center-radius format, specifically `["O", r]`, during the model training phase. As illustrated in Figure 3, regardless of the initial construction logic, the rendering engine eventually computes a deterministic coordinate-level radius to plot the diagram. To improve the stability of structural prediction, we only require the model to regress the integer part of the radius. This design choice effectively reduces the complexity of numerical prediction while preserving the essential geometric structure required for subsequent reasoning.

**Annotations.** The `annotations` field contains visual markers and measurements that are explicitly stated in the problem:

```
"annotations": {
    "right_angles": [
        ["A", "B", "C"],
        ...
    ],
    "length_of_line": [
        [["A", "B"], "5"],
        [["C", "D"], "2*sqrt(3)"],
        ...
    ],
    "measure_of_angle": [
        [["A", "B", "C"], "30"],
        ...
    ]
}
```

- `right_angles`: List of angle triples `["A", "B", "C"]` where $\angle ABC = 90°$.

- `length_of_line`: List of pairs `[[segment], value]` where the segment (two-point list) has the specified length. The value can be a numeric string (e.g., `"5"`) or an expression (e.g., `"2*sqrt(3)"`).

- `measure_of_angle`: List of pairs `[[angle_triple], value]` where the angle (three-point list) has the specified measure in degrees. The value can be a numeric string (e.g., `"30"`) or a radian expression with $\pi$ (e.g., `"pi/6"`).

All annotations must correspond to geometric relations that are *explicitly stated* in the problem text, not inferred through reasoning.

**Quantities.** The `quantities` field contains DSL expressions representing the quantities to be computed or verified:

```
"quantities": [
    "length(A, B)",
    "angle(A, B, C)",
    "length(A, B) - length(C, D)",
    ...
]
```

For computation problems, each expression directly represents a quantity to be found. For proof problems, expressions take the form `left - right` (e.g., `"length(A, B) - length(C, D)"`), which should evaluate to zero if the statement is true.

The complete plotting code structure enables deterministic rendering: given the same plotting code, the renderer produces identical diagrams. This property is crucial for maintaining consistency between the generated problem statement, the

underlying geometric structure, and the visual representation, and enables plotting code to serve as an explicit supervision target for visual–symbolic alignment.

### F.3. Seed Generation

Our seed generation process leverages GenesisGeo (Zhu et al., 2025), an improved implementation of AlphaGeometry and NewClid (Sicca et al., 2024), to construct symbolic geometric structures. To ensure consistency and compatibility with the underlying reasoning system, we maintain the same rule set used by GenesisGeo. These rules form the foundation of our symbolic reasoning engine and enable the construction of dependency graphs from predicate sets.

The rule set we employ is identical to that used in GenesisGeo, ensuring that our generated seeds are compatible with the solver's reasoning capabilities. These rules encode fundamental geometric relationships and theorems, allowing the system to perform forward symbolic reasoning and construct proof dependency graphs. By using the same rule set, we ensure that the symbolic structures generated in our pipeline are directly compatible with GenesisGeo's reasoning engine, enabling seamless integration between seed generation and validation.

**List of rules with names:**

```
r00 Perpendiculars give parallel
perp A B C D, perp C D E F, ncoll A B E => para A B E F

r01 Definition of circle
cong O A O B, cong O B O C, cong O C O D => cyclic A B C D

r02 Parallel from inclination
eqangle A B P Q C D P Q => para A B C D

r03 Arc determines internal angles
cyclic A B P Q => eqangle P A P B Q A Q B

r04 Congruent angles are in a circle
eqangle P A P B Q A Q B, ncoll P Q A B => cyclic A B P Q

r05 Same arc same chord
cyclic A B C P Q R, eqangle C A C B R P R Q => cong A B P Q

r06 Base of half triangle
midp E A B, midp F A C => para E F B C

r07 Thales Theorem I
para A B C D, coll O A C, coll O B D => eqratio3 A B C D O O

r08 Right triangles common angle I
perp A B C D, perp E F G H, npara A B E F => eqangle A B E F C D G H

r09 Sum of angles of a triangle
eqangle a b c d m n p q, eqangle c d e f p q r u => eqangle a b e f m n r u

r10 Ratio cancellation
eqratio a b c d m n p q, eqratio c d e f p q r u => eqratio a b e f m n r u

r11 Bisector theorem I
eqratio d b d c a b a c, coll d b c, ncoll a b c => eqangle a b a d a d a c

r12 Bisector theorem II
eqangle a b a d a d a c, coll d b c, ncoll a b c => eqratio d b d c a b a c

r13 Isosceles triangle equal angles
cong O A O B, ncoll O A B => eqangle O A A B A B O B

r14 Equal base angles imply isosceles
eqangle A O A B B A B O, ncoll O A B => cong O A O B
```

```
r15 Arc determines inscribed angles (tangent)
circle O A B C, perp O A A X => eqangle A X A B C A C B

r16 Same arc giving tangent
circle O A B C, eqangle A X A B C A C B => perp O A A X

r17 Central angle vs inscribed angle I
circle O A B C, midp M B C => eqangle A B A C O B O M

r18 Central angle vs inscribed angle II
circle O A B C, coll M B C, eqangle A B A C O B O M => midp M B C

r19 Hypothenuse is diameter
perp A B B C, midp M A C => cong A M B M

r20 Diameter is hypotenuse
circle O A B C, coll O A C => perp A B B C

r21 Cyclic trapezoid
cyclic A B C D, para A B C D => eqangle A D C D C D C B

r22 Bisector Construction
midp M A B, perp O M A B => cong O A O B

r23 Bisector is perpendicular
cong A P B P, cong A Q B Q => perp A B P Q

r24 Cyclic kite
cong A P B P, cong A Q B Q, cyclic A B P Q => perp P A A Q

r25 Diagonals of parallelogram I
midp M A B, midp M C D => para A C B D

r26 Diagonals of parallelogram II
midp M A B, para A C B D, para A D B C => midp M C D

r27 Thales theorem II
eqratio O A A C O B B D, coll O A C, coll O B D, ncoll A B C, sameside A O C B O D => para
    A B C D

r28 Overlapping parallels
para A B A C => coll A B C

r29 Midpoint is an eqratio
midp M A B, midp N C D => eqratio M A A B N C C D

r30 Right triangles common angle II
eqangle A B P Q C D U V, perp P Q U V => perp A B C D

r31 Denominator cancelling
eqratio A B P Q C D U V, cong P Q U V => cong A B C D

r34 AA Similarity of triangles (direct)
eqangle B A B C Q P Q R, eqangle C A C B R P R Q, ncoll A B C, sameclock A B C P Q R =>
    simtri A B C P Q R

r35 AA Similarity of triangles (reverse)
eqangle B A B C Q R Q P, eqangle C A C B R Q R P, ncoll A B C, sameclock A B C P R Q =>
    simtrir A B C P Q R

r36 ASA Congruence of triangles (direct)
eqangle B A B C Q P Q R, eqangle C A C B R P R Q, ncoll A B C, cong A B P Q, sameclock A B
    C P Q R => contri A B C P Q R

r37 ASA Congruence of triangles (reverse)
```

```
eqangle B A B C Q R Q P, eqangle C A C B R Q R P, ncoll A B C, cong A B P Q, sameclock A B
    C P R Q => contrir A B C P Q R
```

r41 Thales theorem III
```
para a b c d, coll m a d, coll n b c, eqratio m a m d n b n c, sameside m a d n b c =>
    para m n a b
```

r42 Thales theorem IV
```
para a b c d, coll m a d, coll n b c, para m n a b => eqratio m a m d n b n c
```

r43 Orthocenter theorem
```
perp a b c d, perp a c b d => perp a d b c
```

r44 Pappus's theorem
```
coll a b c, coll p q r, coll x a q, coll x p b, coll y a r, coll y p c, coll z b r, coll z
    c q => coll x y z
```

r45 Simson's line theorem
```
cyclic a b c p, coll a l c, perp p l a c, coll m b c, perp p m b c, coll n a b, perp p n a
    b => coll l m n
```

r46 Incenter theorem
```
eqangle a b a x a x a c, eqangle b a b x b x b c, ncoll a b c => eqangle c b c x c x c a
```

r47 Circumcenter theorem
```
midp m a b, perp x m a b, midp n b c, perp x n b c, midp p c a => perp x p c a
```

r48 Centroid theorem
```
midp m a b, coll m x c, midp n b c, coll n x a, midp p c a => coll x p b
```

r49 Recognize center of cyclic (circle)
```
circle O A B C, cyclic A B C D => cong O A O D
```

r50 Recognize center of cyclic (cong)
```
cyclic A B C D, cong O A O B, cong O C O D, npara A B C D => cong O A O C
```

r51 Midpoint splits in two
```
midp M A B => rconst M A A B 1/2
```

r52 Properties of similar triangles (Direct)
```
simtri A B C P Q R => eqangle B A B C Q P Q R, eqratio B A B C Q P Q R
```

r53 Properties of similar triangles (Reverse)
```
simtrir A B C P Q R => eqangle B A B C Q R Q P, eqratio B A B C Q P Q R
```

r54 Definition of midpoint
```
cong M A M B, coll M A B => midp M A B
```

r55 Properties of midpoint (cong)
```
midp M A B => cong M A M B
```

r56 Properties of midpoint (coll)
```
midp M A B => coll M A B
```

r57 Pythagoras theorem
```
PythagoreanPremises a b c => PythagoreanConclusions a b c
```

r58 Same chord same arc I
```
cyclic a b c p q r, cong a b p q, sameclock c a b r p q, sameside c a b r p q => eqangle c
    a c b r p r q
```

r59 Same chord same arc II
```
cyclic a b c p q r, cong a b p q, sameclock c b a r p q, nsameside c b a r p q => eqangle
    c a c b r q r p
```

```
r60 SSS Similarity of triangles (Direct)
eqratio B A B C Q P Q R, eqratio C A C B R P R Q, ncoll A B C, sameclock A B C P Q R =>
    simtri A B C P Q R

r61 SSS Similarity of triangles (Reverse)
eqratio B A B C Q P Q R, eqratio C A C B R P R Q, ncoll A B C, sameclock A B C P R Q =>
    simtrir A B C P Q R

r62 SAS Similarity of triangles (Direct)
eqratio B A B C Q P Q R, eqangle B A B C Q P Q R, ncoll A B C, sameclock A B C P Q R =>
    simtri A B C P Q R

r63 SAS Similarity of triangles (Reverse)
eqratio B A B C Q P Q R, eqangle B A B C Q P Q R, ncoll A B C, sameclock A B C P R Q =>
    simtrir A B C P Q R

r64 SSS Congruence of triangles (Direct)
cong A B P Q, cong B C Q R, cong C A R P, ncoll A B C, sameclock A B C P Q R => contri A B
    C P Q R

r65 SSS Congruence of triangles (Reverse)
cong A B P Q, cong B C Q R, cong C A R P, ncoll A B C, sameclock A B C P R Q => contrir A
    B C P Q R

r66 SAS Congruence of triangles (Direct)
cong A B P Q, cong B C Q R, eqangle B A B C Q P Q R, ncoll A B C, sameclock A B C P Q R =>
    contri A B C P Q R

r67 SAS Congruence of triangles (Reverse)
cong A B P Q, cong B C Q R, eqangle B A B C Q P Q R, ncoll A B C, sameclock A B C P R Q =>
    contrir A B C P Q R

r68 Similarity without scaling (Direct)
eqratio B A B C Q P Q R, eqratio C A C B R P R Q, ncoll A B C, cong A B P Q, sameclock A B
    C P Q R => contri A B C P Q R

r69 Similarity without scaling (Reverse)
eqratio B A B C Q P Q R, eqratio C A C B R P R Q, ncoll A B C, cong A B P Q, sameclock A B
    C P R Q => contrir A B C P Q R
```

## G. API Usage

In section 4.3, we evaluated the difficulty of our generated problems using closed-source large language model APIs. Specifically, we accessed Google's Gemini-2.5-Pro and OpenAI's GPT-5 through the OpenRouter API, while Qwen3-VL-32B-Thinking was served locally. Throughout the evaluation process, we ensured that both the system prompt and user prompt remained consistent across all models to guarantee fair and correct assessment.

## H. Limitations

Although our method successfully synthesizes geometric data, it still has several limitations.

**Dependence on LLM Reasoning Capabilities.** First, during instantiation, our approach heavily relies on the reasoning capabilities of existing LLMs. We use GPT-OSS-120B as the reasoning model, and approximately 50% of samples are discarded during this process. We believe that using stronger reasoning models in the future can improve this efficiency. Additionally, methods relying on LLMs inevitably face hallucination issues, and validating their results requires additional computational overhead.

**Model-Specific Evaluation.** Second, our experiments are primarily conducted on the Qwen series of models, such as Qwen2.5VL and Qwen3VL. While our approach can theoretically be directly transferred to other VLM backbones, we have not extensively validated this transferability. We also look forward to our data being used for training other series of VLMs.

**Geometric Scope and Visual Style.** Finally, our constructed geometry is limited to planar Euclidean geometry. This limitation stems from the structural foundation generated during the Seed Generation stage, which is inherently planar. Our work does not cover higher-dimensional geometric types, which represents a direction for future research. Additionally, the images we generate are idealized and presented in a textbook style, and therefore do not cover handwritten, sketch, or real exam paper styles.

# I. Prompts

---

### LLM Generator Prompt (Computation)

```
You are a professional geometry problem authoring expert, proficient in
Euclidean plane geometry and competition-level geometry problems (such as
AMC / AIME / Math Olympiad).

[Task Objective]

Based on the input <problem> (geometric premises) and conclusion, you need to:
1. Understand the geometric configuration described (points, line segments,
   perpendicular, parallel, equal lengths, similarity, collinearity, etc.).
2. On the basis of this geometric configuration, design a high-quality,
   high-difficulty numerical geometry problem (with only one solving target).
3. Provide a complete problem statement, detailed reasoning process (CoT),
   and the final numerical answer.

The problem statement should use "geometric language" only, do not use
coordinate systems.

-------------------------------
[Problem Requirements]

1. **Numerical Settings**
   - Set reasonable numerical values for side lengths, angles, ratios, etc.
     (recommended: integers or simple fractions in the range 2-20, e.g.,
     3/2, 5/3).
   - The answer should not be too simple (avoid obvious results like 0, 1, 2;
     avoid cases where the answer is identical to a given quantity).

2. **Solving Target Type (only one)**
   The problem should end with a single solving target, ending with "Find...".
   It can be:
   - The length of a line segment;
   - The degree measure of an angle or its trigonometric value;
   - The ratio of two line segments;
   - The area / perimeter of a triangle or quadrilateral;
   - Angles, arc lengths, areas involving circles;
   - Arithmetic operations on the above quantities (e.g., AB² + AC², area ratios).

3. **Geometric Logic Requirements**
   - The problem must be logically self-consistent, non-contradictory, and
     have a unique solution.
   - Degenerate cases are not allowed (e.g., three collinear points forming
     a triangle).
   - If new points are introduced (e.g., intersection points, foots of
     perpendiculars, midpoints), they must be clearly defined.

4. **Problem Statement Format**
   - Natural language narrative, using LaTeX mathematical notation
     (e.g., AB, angleABC, S_triangleABC).
   - Do not include reasoning content like conclusion, only include
     construction content.
   - The last sentence should end with "Find...".
```

```
5. **CoT Reasoning Process**
   - Write in Step1 / Step2 / Step3... format.
   - Use LaTeX mathematical notation and formulas.
   - Do not mention DSL, <problem>, symbolic propositions, etc., only reason
     based on the problem statement.

6. **Answer Format**
   - "answer": Only provide the final numerical value or expression,
     e.g., "15/2", "5sqrt3".
   - Do not include explanatory text.

-------------------------------
[Input Format]

<problem> {problem} </problem>
<conclusion> {conclusion} </conclusion>
{aux_section}

[Output Format]

Output only a JSON object:

{
  "question": "(Problem statement using LaTeX math notation)",
  "cot": "Step1 ...\nStep2 ...\nStep3 ...",
  "answer": "(Final numerical result, only provide value or expression)"
}
```

## LLM Generator Prompt (Proof)

```
You are a professional geometry problem authoring expert, proficient in
Euclidean plane geometry and competition-level geometry problems (such as
AMC / AIME / Math Olympiad).

[Task Objective]

Based on the input <problem> (geometric premises) and <conclusion> (the
conclusion to be proved), you need to:
1. Understand the geometric configuration described (points, line segments,
   perpendicular, parallel, equal lengths, similarity, collinearity, etc.).
2. Convert the symbolic geometric premises and conclusion into a natural
   language geometry proof problem.
3. Provide a complete problem statement, detailed proof process (CoT), and
   proof conclusion.

The problem statement should use "geometric language" only, do not use
coordinate systems.

------------------------------
[Problem Requirements]

1. **Problem Statement Construction**
   - Convert the premises in <problem> into natural language descriptions.
   - Convert the conclusion in <conclusion> into a natural language proof goal.
   - You may add auxiliary conditions in the problem statement (such as
     midpoints, angle bisectors, division points, etc.), but they must be
     reasonable and compatible with the geometric structure in <problem>.
   - Do not introduce new "unknown quantities" or additional parameters
     (such as new variables x, y, t, k, etc.) in the problem statement and
     proof process. If quantity relationships need to be expressed, use
     known quantities (side lengths, angles, ratios given in the problem)
     or their combinations directly.

2. **Proof Goal**
   - The problem should end with "Prove that..." to clearly state the
     geometric conclusion to be proved.
   - The proof goal can be:
     - Two line segments are equal or proportional;
     - Two angles are equal or proportional;
     - Certain points are collinear or concyclic;
     - Certain lines are parallel or perpendicular;
     - Certain triangles are similar or congruent;
     - Other geometric relationships.

3. **Geometric Logic Requirements**
   - The problem must be logically self-consistent, non-contradictory, and
     provable.
   - Degenerate cases are not allowed (e.g., three collinear points forming
     a triangle).
   - If new points are introduced (e.g., intersection points, foots of
     perpendiculars, midpoints), they must be clearly defined.
   - The premises must be sufficient to derive the conclusion.

4. **Problem Statement Format**
   - Natural language narrative in Chinese, using LaTeX mathematical notation
     (e.g., AB, angleABC, S_triangleABC).
   - Do not include reasoning content like conclusion, only include
     construction content and proof goal.
   - The last sentence should end with "Prove that...".

5. **CoT Proof Process**
```

```
    – Write the complete proof process in Step1 / Step2 / Step3... format.
    – Use LaTeX mathematical notation and formulas.
    – Each step should explain the geometric theorem or property used
      (such as congruence, similarity, parallel line properties, circle
      properties, etc.).
    – Do not mention DSL, <problem>, symbolic propositions, etc., only
      perform geometric reasoning based on the problem statement.
    – The proof process should be logically clear and complete.

6. **Answer Format**
    – "answer": Provide the proof conclusion, formatted as "QED" or
      "Therefore, [conclusion]".
    – If the conclusion involves quantity relationships, express the conclusion
      directly using known quantities or their combinations, do not introduce
      additional unknown quantities.

------------------------------
[Input Format]

<problem> {problem} </problem>
<conclusion> {conclusion} </conclusion>
{aux_section}

[Output Format]

Output only a JSON object:

{
  "question": "(Problem statement using LaTeX math notation, ending with
               'Prove that...')",
  "cot": "Step1 ...\nStep2 ...\nStep3 ...",
  "answer": "(Proof conclusion, e.g., 'QED' or 'Therefore, [conclusion]')"
}
```

## LLM Plotter Prompt (Coordinates)

```
Your task is: Generate a set of 2D numerical coordinates for all points
appearing in the problem that satisfy all geometric relationships in the
problem statement, making the figure structure reasonable, clear, and stable.

====================
[Problem Text]
{question}
====================

Please output a piece of **executable Python code** that strictly satisfies
the following requirements:

=======================================================
[Global Code Constraints]
=======================================================
1. You must generate **complete, executable Python code**, do not output
   explanations, comments, or extra text.
2. All variables, functions, and constants must be explicitly defined or
   imported in the code.
3. All coordinates must be **concrete numerical values (float or int)**,
   do not use symbolic expressions, lambdas, undefined variables, or lazy
   expressions.
4. Standard library imports are allowed (e.g., `import math`, `import numpy`),
   but file I/O and network requests are prohibited.

=======================================================
[Geometric Construction Rules]
=======================================================
1. **Point Name Restrictions**: Only use points mentioned in the problem
   statement, do not introduce any new points.
2. **Geometric Relationship Constraints**: Coordinates must satisfy all
   geometric relationships appearing in the problem statement, for example:
   - Perpendicular, parallel
   - Collinear
   - Midpoint
   - On circle
   - Ratio relationships
   - Equal lengths
   - ...
   Do not add properties not mentioned in the problem (such as isosceles,
   perpendicular, parallel, etc.).
3. **Non-degenerate Figure**:
   - Do not allow two different points to coincide
   - Do not allow three points that should form an angle/triangle to be
     nearly collinear
   - Coordinates should be dispersed and stable
4. Geometric relationships must be guaranteed to hold through explicit
   construction (e.g., through slope, vectors, distance from point to
   circle center, etc.).

=======================================================
[Final Output Structure]
=======================================================
At the end of the code, construct and print the following dictionary. The
dictionary should only contain point coordinates, do not include any other
content (field names must match exactly):

result = {
    "points": {   # Coordinate mapping for all points
        "A": (xA, yA),
        "B": (xB, yB),
```

```
        ...
    }
}

import json
print(json.dumps(result, ensure_ascii=False))

You must output only Python code, do not include any explanatory text or
non-code content.
```

## LLM Plotter Prompt (Annotations)

```
Your task is: Extract annotation information (annotations), line segments
(segments), and circles (circles) based only on "conditions directly stated
in text" in the problem statement. Do not add any implicitly inferred
information.

====================
[Problem Text]
{question}
====================

Please directly output a JSON object containing the following fields:

{
    "segments": [
        ["A", "B"],
        ["B", "C"],
        ...
    ],
    "circles": [
        ["C1", "O", 5],
        ["C2", "A", "B", "diameter"],
        ["C3", "A", "B", "C"],
        ...
    ],
    "annotations": {
        "right_angles": [
            ["A", "B", "C"],
            ...
        ],
        "length_of_line": [
            [["A", "B"], "5"],
            [["C", "D"], "2*sqrt(3)"],
            ...
        ],
        "measure_of_angle": [
            [["A", "B", "C"], "30"],
            ...
        ]
    }
}

[Extraction Rules]

1. segments (line segments):
   - If the problem describes a polygon (e.g., triangle ABC, quadrilateral
     ABCD, etc.), you must include all edges of the polygon in segments,
     do not miss any edges.
   - If the problem implicitly mentions an edge, also include it in segments.
   - If the problem explicitly mentions a line segment or edge (e.g.,
     "connect AD", "draw BE"), the corresponding segment should also appear
     in segments.
   - Segment format: ["A", "B"] represents a line segment with endpoints
     A and B.

2. circles:
   - Unless the problem **explicitly mentions "circle", "inscribed circle",
     "circumscribed circle" or similar descriptions**, circles must be an
     empty array [].
   - If the problem involves circles, each circle must use a **unique circle
     ID**: C1, C2, C3, ...
   - Circle formats:
```

```
        ["C1", "O", 5]                  # Circle C1, center O, radius 5
        ["C2", "O", "P"]                # Circle C2, center O, OP as radius
        ["C3", "A", "B", "diameter"]    # Circle C3, AB as diameter
        ["C4", "A", "B", "C"]           # Circle C4, passing through A, B, C
```

3. annotations:
   Only annotate geometric quantities **directly stated in the problem text**,
   do not add information inferred through reasoning.

   Allowed annotation examples:
   - Angles and lengths should only be annotated with numbers.
   - If the problem states "angleABC = 30degrees", you can record in `measure_of_angle`:
     [["A", "B", "C"], "30"]
   - If the problem states "AB = 5" or "AB = 2sqrt3", you can record in
     `length_of_line`: [["A", "B"], "5"] or [["A", "B"], "2*sqrt(3)"]
   - If the problem states "angleABC is a right angle" or "angleABC = 90degrees", you can
     record in `right_angles`: ["A", "B", "C"]

   Explicitly prohibited:
   - Do not annotate information "inferred" from geometric properties, for
     example:
     - If a triangle is isosceles, do not add "base angles are equal";
     - If a point is a midpoint, do not add "two segments are equal";
     - If some edges are parallel, do not add "corresponding angles are equal", etc.
     - If it's a square, do not annotate four right angles.
   - If the problem does not directly give numerical or right angle information,
     do not annotate, **prefer less than wrong**.

   Format requirements:
   - `right_angles`: Elements are three-point lists ["A", "B", "C"],
     representing angleABC is a right angle.
   - `length_of_line`: Elements are [ ["A", "B"], "expression string" ].
     - Use `sqrt(x)` form for square roots in expressions, e.g., "2*sqrt(3)",
       do not use `math.sqrt(3)`.
     - For simple integers or fractions, you can directly write strings
       "5", "3/2", etc.
   - `measure_of_angle`: Elements are [ ["A", "B", "C"], "angle measure
     (default unit is degrees)" ], e.g., "30".
   - If a certain type of information does not appear in the problem at all,
     set the corresponding field to an empty array [].

Please directly output JSON, do not include any other explanatory text.

## LLM Plotter Prompt (Quantities - Computation)

```
Your task is: Based on "what the problem asks you to find" in the problem,
generate DSL expressions in the quantities list (do not calculate, do not
output specific numerical answers).

=====================
[Problem Text]
{question}
=====================

Please directly output a JSON object in the following format:

{
    "quantities": [
        "length(A, B)",
        "angle(A, B, C)",
        ...
    ]
}

[quantities Generation Rules (extremely important)]

1. Each element in the quantities list must be a string containing a **DSL
   expression**, representing "what the problem asks you to find". **Only
   write expressions, not numerical values**.
2. Only the following DSL forms are allowed, do not invent new function
   names or syntax. Special note: The first parameter of circle-related
   functions must be a circle ID (e.g., C1), **do not write point names
   (e.g., E)**, for example:
   - Correct: `radius(C1)`
   - Wrong: `radius(E)` (E is a point name, not a circle ID)

(1) Point-related geometric quantities:
    length(A, B)              # Find the length of line segment AB
    angle(A, B, C)            # Find the measure of angleABC (default unit is degrees)
    tan(A, B, C)              # Find the tangent value of angleABC
    sin(A, B, C)              # Find the sine value of angleABC
    cos(A, B, C)              # Find the cosine value of angleABC
    area(A, B, C, D, ...)     # Find the area of polygon A-B-C-D-...
    perimeter(A, B, C, D, ...)# Find the perimeter of polygon A-B-C-D-...

(2) Angles and trigonometric functions between two lines:
    angle_between_lines(A, B, C, D)  # Find the angle measure between lines
                                     # AB and CD (0degrees~90degrees)
    tan_between_lines(A, B, C, D)    # Find the tangent of the angle between
                                     # lines AB and CD
    sin_between_lines(A, B, C, D)    # Find the sine of the angle between
                                     # lines AB and CD
    cos_between_lines(A, B, C, D)    # Find the cosine of the angle between
                                     # lines AB and CD
    # Where AB represents the first line, CD represents the second line;
    # A, B, C, D are all points appearing in the problem.

(3) Circle-related quantities (must explicitly write circle ID as the first
    parameter, e.g., C1, C2, do not use circle center point names or other
    symbols):
    central_angle(C1, A, B)          # Find the central angle measure in circle
                                     # C1 with center as vertex, passing
                                     # through arc AB
    arc_length(C1, A, B)             # Find the length of arc AB on circle C1
    sector_area(C1, A, B)            # Find the sector area in circle C1
                                     # corresponding to arc AB
```

```
    arc_inscribed_angle(C1, A, B)   # Find the inscribed angle measure in
                                     # circle C1 corresponding to arc AB
    circle_area(C1)                 # Find the area of circle C1
    circle_perimeter(C1)            # Find the perimeter of circle C1
    segment_area(C1, A, B)          # Find the segment area in circle C1
                                     # corresponding to chord AB
    radius(C1)                      # Find the radius of circle C1
    diameter(C1)                    # Find the diameter of circle C1
```

3. **Usually, the problem asks for only one quantity, so quantities should contain only one expression**; if the problem asks for multiple quantities, you can write multiple expressions in quantities.
4. Do not calculate the values of these expressions, and do not write numerical results into quantities, only write DSL expressions as strings.
5. Simple arithmetic operations are allowed in a string (if the problem indeed requires it), for example:
    "length(A, B) +-*/ length(B, C)"
   But they must use the above basic predicates as atomic units, and the form must be clear and easy to parse.
6. **If the problem asks for a square (e.g., "find AB²", "find the square of AB"), it can be expressed as a product form**, for example:
    "length(A, B) * length(A, B)"    # Represents the square of AB
   Or use multiplication operators to represent the square relationship.

Please directly output JSON, do not include any other explanatory text.

## LLM Plotter Prompt (Quantities - Proof)

```
Your task is: Based on the "prove" conclusion in the problem, generate DSL
expressions in the quantities list for numerical verification.

=====================
[Problem Text]
{question}
=====================

Please directly output a JSON object in the following format:

{
    "quantities": [
        "length(A, B) - length(C, D)",
        "angle(A, B, C) - angle(D, E, F)",
        ...
    ]
}

[quantities Generation Rules (for proof problems, extremely important)]

1. Each element in the quantities list must be a string containing a **DSL
   expression**, representing an equality relationship to be verified.
2. **Key points**:
   - For proof problems, you need to convert the "prove" conclusion into
     **equality verification expressions** in the format `left - right`,
     expecting the result to be 0;
   - **The first parameter of all circle-related functions (such as
     central_angle / arc_length / sector_area / arc_inscribed_angle /
     circle_area / circle_perimeter / segment_area / radius / diameter)
     must be a circle ID (e.g., C1, C2), do not write circle center point
     names or other symbols**.

   Examples:
   - If the conclusion is "AB = CD", write: `"length(A, B) - length(C, D)"`
   - If the conclusion is "angleABC = angleDEF", write:
     `"angle(A, B, C) - angle(D, E, F)"`
   - If the conclusion is "AB:CD = EF:GH", write:
     `"length(A, B) / length(C, D) - length(E, F) / length(G, H)"`
   - If the conclusion is "AB parallel CD" (parallel), write:
     `"angle_between_lines(A, B, C, D) - 0"`
   - If the conclusion is "AB perpendicular CD" (perpendicular), write:
     `"angle_between_lines(A, B, C, D) - 90"`
   - If the conclusion is "points A, B, C are collinear", write:
     `"area(A, B, C) - 0"`

3. Only the following DSL forms are allowed, do not invent new function names
   or syntax. Special note: The first parameter of circle-related functions
   must be a circle ID (e.g., C1), **do not write point names (e.g., E)**,
   for example:
   - Correct: `radius(C1)`
   - Wrong: `radius(E)` (E is a point name, not a circle ID)

(1) Point-related geometric quantities:
    length(A, B)             # Find the length of line segment AB
    angle(A, B, C)           # Find the measure of angleABC (default unit is degrees)
    tan(A, B, C)             # Find the tangent value of angleABC
    sin(A, B, C)             # Find the sine value of angleABC
    cos(A, B, C)             # Find the cosine value of angleABC
    area(A, B, C, D, ...)    # Find the area of polygon A-B-C-D-...
    perimeter(A, B, C, D, ...)# Find the perimeter of polygon A-B-C-D-...
```

```
(2) Angles and trigonometric functions between two lines:
    angle_between_lines(A, B, C, D)  # Find the angle measure between lines
                                      # AB and CD (0degrees~90degrees)
    tan_between_lines(A, B, C, D)    # Find the tangent of the angle between
                                      # lines AB and CD
    sin_between_lines(A, B, C, D)    # Find the sine of the angle between
                                      # lines AB and CD
    cos_between_lines(A, B, C, D)    # Find the cosine of the angle between
                                      # lines AB and CD
    # Where AB represents the first line, CD represents the second line;
    # A, B, C, D are all points appearing in the problem.

(3) Circle-related quantities (must explicitly write circle ID as the first
    parameter, e.g., C1, C2, do not use circle center point names or other
    symbols):
    central_angle(C1, A, B)         # Find the central angle measure in circle
                                     # C1 with center as vertex, passing
                                     # through arc AB
    arc_length(C1, A, B)            # Find the length of arc AB on circle C1
    sector_area(C1, A, B)           # Find the sector area in circle C1
                                     # corresponding to arc AB
    arc_inscribed_angle(C1, A, B)   # Find the inscribed angle measure in
                                     # circle C1 corresponding to arc AB
    circle_area(C1)                 # Find the area of circle C1
    circle_perimeter(C1)            # Find the perimeter of circle C1
    segment_area(C1, A, B)          # Find the segment area in circle C1
                                     # corresponding to chord AB
    radius(C1)                      # Find the radius of circle C1
    diameter(C1)                    # Find the diameter of circle C1

3. If the conclusion involves multiple equality relationships, you can write
   multiple expressions in quantities, each expression corresponding to an
   equality to be verified.
4. Do not calculate the values of these expressions, and do not write
   numerical results into quantities, only write DSL expressions as strings.
5. Simple arithmetic operations are allowed in a string, for example:
       "length(A, B) - length(C, D)"
       "angle(A, B, C) - angle(D, E, F)"
       "length(A, B) / length(C, D) - length(E, F) / length(G, H)"
   But they must use the above basic predicates as atomic units, and the
   form must be clear and easy to parse.

Please directly output JSON, do not include any other explanatory text.
```

## LLM Plotter Prompt (Unified)

You now have **three tasks** that need to be completed in **one complete, executable Python code** (do not split into multiple code segments):

Task1 (Geometric Construction):
    Generate 2D numerical coordinates for all points appearing in the problem
    that satisfy the geometric relationships in the problem statement, and
    provide necessary line segment / circle information, making the figure
    structure reasonable and stable.

Task2 (Plain Text Annotations):
    Generate annotations based only on "conditions directly stated in text"
    in the problem statement (do not add any implicitly inferred information).

Task3 (Quantities DSL):
    Based on "what the problem asks you to find" in the problem, only generate
    DSL expressions in the quantities list (do not calculate, do not output
    specific numerical answers).

====================
[Problem Text]
{question}
====================

Please output a piece of **executable Python code** that strictly satisfies
the following requirements:

[Global Code Constraints]
1. The code must have no syntax errors and can be directly executed in a
   Python interpreter.
2. All variables, constants, and functions used must be defined or imported
   beforehand, do not use undefined names.
3. You can use any legal method for numerical calculations (including math,
   fractions, numpy, etc.), but:
   - All point coordinates and circle radii written to result must be
     numerical values (int or float), **cannot** be symbolic expressions or
     lazy expressions.
4. The code **prohibits calculating the final answer required by the problem**,
   and **prohibits printing or outputting any answer numerical values**.
   quantities can only be DSL expressions in string form.
5. Standard library imports are allowed (e.g., `import math`, `import json`),
   but file I/O and network requests are prohibited.

[Geometric Construction Rules (Task1)]

1. Points (points)
   - Only use point names appearing in the problem statement (e.g., A, B, C,
     D, O, etc.), **do not introduce any new points**.
   - Each point's coordinates must be in numerical form `(x, y)`, where x, y
     are int or float.
   - Do not allow two different points to coincide, i.e., different points
     must have different coordinates.
   - Coordinates can be chosen arbitrarily, but must ensure they satisfy the
     geometric relationships in the problem statement (such as perpendicular,
     parallel, collinear, on circle, etc.).

2. Line Segments (segments)
   - The segment list is used to describe the basic edge structure of the figure.
   - If the problem describes a polygon (e.g., triangle ABC, quadrilateral
     ABCD, etc.), you must include all edges of the polygon in segments,

```
        do not miss any edges.
      - Segments are stored as tuples ("A", "B"), representing a line segment
        with endpoints A and B.
      - If the problem explicitly mentions a line segment or edge (e.g.,
        "connect AD", "draw BE"), the corresponding segment should also appear
        in segments, as long as it does not conflict with "prohibiting
        introducing new points".

  3. Circles (circles)
      - Unless the problem **explicitly mentions "circle", "inscribed circle",
        "circumscribed circle" or similar descriptions**, then:
        - `circles` must be an empty list `[]`.
      - If the problem involves circles, each circle must use a **unique circle
        ID**: C1, C2, C3, ...
      - Allowed circle representation formats (the first parameter must be a
        circle ID string):
            ["C1", "O", 5]                # Circle C1, center O, radius 5
            ["C2", "O", "P"]              # Circle C2, center O, OP as radius
            ["C3", "A", "B", "diameter"]  # Circle C3, AB as diameter
            ["C4", "A", "B", "C"]         # Circle C4, passing through A, B, C
      - If the problem mentions "inscribed circle", "circle determined by three
        points" and similar situations, prefer using three-point circle or
        diameter circle forms.
      - Do not generate additional circles not mentioned in the problem.

  [annotations Generation Rules (Task2)]

  Only annotate geometric quantities **directly stated in the problem text**,
  do not add information inferred through reasoning. Including but not limited to:

  1. Allowed annotation examples:
      - If the problem states "angleABC = 30degrees", you can record in `measure_of_angle`:
          [["A", "B", "C"], "30"]
      - If the problem states "AB = 5" or "AB = 2sqrt3", you can record in
        `length_of_line`:
          [["A", "B"], "5"] or [["A", "B"], "2*sqrt(3)"]
      - If the problem states "angleABC is a right angle" or "angleABC = 90degrees", you can
        record in `right_angles`:
          ["A", "B", "C"]

  2. Explicitly prohibited:
      - Do not annotate information "inferred" from geometric properties, for
        example:
        - If a triangle is isosceles, do not add "base angles are equal";
        - If a point is a midpoint, do not add "two segments are equal";
        - If some edges are parallel, do not add "corresponding angles are equal", etc.
      - If the problem does not directly give numerical or right angle information,
        do not annotate, **prefer less than wrong**.

  3. Format requirements:
      - `right_angles`: Elements are three-point lists ["A", "B", "C"],
        representing angleABC is a right angle.
      - `length_of_line`: Elements are [ ["A", "B"], "expression string" ].
        - Use `sqrt(x)` form for square roots in expressions, e.g., "2*sqrt(3)",
          do not use `math.sqrt(3)`.
        - For simple integers or fractions, you can directly write strings
          "5", "3/2", etc.
      - `measure_of_angle`: Elements are [ ["A", "B", "C"], "angle measure
        (default unit is degrees)" ], e.g., "30".
      - If a certain type of information does not appear in the problem at all,
        set the corresponding field to an empty list `[]`.
```

```
4. Do not annotate decimal lengths or angles (i.e., decimals not explicitly
   given in the problem, do not add extra annotations).

[quantities Generation Rules (Task3, extremely important)]

1. Each element in the quantities list must be a string containing a **DSL
   expression**, representing "what the problem asks you to find". **Only
   write expressions, not numerical values**.
2. Only the following DSL forms are allowed, do not invent new function names
   or syntax.

(1) Point-related geometric quantities:
    length(A, B)              # Find the length of line segment AB
    angle(A, B, C)            # Find the measure of angleABC (default unit is degrees)
    tan(A, B, C)              # Find the tangent value of angleABC
    sin(A, B, C)              # Find the sine value of angleABC
    cos(A, B, C)              # Find the cosine value of angleABC
    area(A, B, C, D, ...)     # Find the area of polygon A-B-C-D-...
    perimeter(A, B, C, D, ...)# Find the perimeter of polygon A-B-C-D-...

(2) Angles and trigonometric functions between two lines (new):
    angle_between_lines(A, B, C, D)  # Find the angle measure between lines
                                     # AB and CD (0degrees~90degrees)
    tan_between_lines(A, B, C, D)    # Find the tangent of the angle between
                                     # lines AB and CD
    sin_between_lines(A, B, C, D)    # Find the sine of the angle between
                                     # lines AB and CD
    cos_between_lines(A, B, C, D)    # Find the cosine of the angle between
                                     # lines AB and CD
    # Where AB represents the first line, CD represents the second line;
    # A, B, C, D are all points appearing in the problem.

(3) Circle-related quantities (must explicitly write circle ID as the first
    parameter):
    central_angle(C1, A, B)       # Find the central angle measure in circle
                                  # C1 with center as vertex, passing
                                  # through arc AB
    arc_length(C1, A, B)          # Find the length of arc AB on circle C1
    sector_area(C1, A, B)         # Find the sector area in circle C1
                                  # corresponding to arc AB
    arc_inscribed_angle(C1, A, B) # Find the inscribed angle measure in
                                  # circle C1 corresponding to arc AB
    circle_area(C1)               # Find the area of circle C1
    circle_perimeter(C1)          # Find the perimeter of circle C1
    segment_area(C1, A, B)        # Find the segment area in circle C1
                                  # corresponding to chord AB
    radius(C1)                    # Find the radius of circle C1
    diameter(C1)                  # Find the diameter of circle C1

3. If the problem asks for only one quantity, quantities should contain only
   one expression; if the problem asks for multiple quantities, you can write
   multiple expressions in quantities.
4. Do not directly calculate the values of these expressions in the code,
   and do not write numerical results into quantities, only write DSL
   expressions as strings.
5. Simple arithmetic operations are allowed in a string (if the problem
   indeed requires it), for example:
       "length(A, B) + length(B, C)"
   But they must use the above basic predicates as atomic units, and the
   form must be clear and easy to parse.
```

```
[Final Output Structure]

Please construct and print the following dictionary object at the end of the
code (field names and structure must match exactly):

result = {
    "points": {  # Coordinate mapping for all points
        "A": (xA, yA),
        "B": (xB, yB),
        # ...
    },
    "segments": [
        ("A", "B"),
        ("B", "C"),
        # ...
    ],
    "circles": [
        # Each circle must use a Ci ID
        # ["C1", "O", 5],
        # ["C2", "A", "B", "diameter"],
        # ["C3", "A", "B", "C"],
    ],
    "quantities": [
        # Only DSL expression strings appear, no numerical answers
        # "length(A, B)",
        # "angle(A, B, C)",
        # "angle_between_lines(A, B, C, D)",
        # "central_angle(C1, A, B)",
        # ...
    ],
    "annotations": {
        "right_angles": [
            ["A", "B", "C"],
            # ...
        ],
        "length_of_line": [
            [["A", "B"], "5"],
            [["C", "D"], "2*sqrt(3)"],
            # ...
        ],
        "measure_of_angle": [
            [["A", "B", "C"], "30"],
            # ...
        ]
    },
}

import json
print(json.dumps(result, ensure_ascii=False))
```

## LLM Judge Prompt

```
You are a professional geometry problem correctness verification expert.

Your task: Determine whether the problem is **mathematically self-consistent**,
**conditions are complete**, and **solvable**.

Input as follows:

Problem: {question}

Solution process: {cot}
Answer: {answer}

[Evaluation Criteria]

Please judge based only on the following two points:
1. **Correctness**: Is the problem description self-consistent, with no
   contradictions or logical conflicts?
2. **Solvability**: Are the conditions sufficient to derive the answer? And
   determine whether cot and answer are consistent with the problem.

[Output]

Please output strict JSON, do not add any extra text:

{
  "passed": true/false,
  "reason": "1-2 sentences explaining the core reason"
}
```

## LLM Judge Prompt (Plotting Code)

```
Evaluate the quality of plotting code. Check: DSL correctness, completeness,
consistency, drawability, reasonableness.

Problem: {question}

Plotting code:
{
  "dsl": "{dsl}",
  "segments": {segments_json},
  "circles": {circles_json}
}

Output JSON (only JSON, no other text):
{
  "passed": true/false,
  "reason": "Evaluation reason (brief explanation, 1-2 sentences)",
  "score": 0-100
}
```

## VisualizeQA Prompt (Step 1: Text Simplification)

You are a professional mathematics editor. Your task is to rewrite a geometry
problem statement into a **concise, natural, and fluent** version. Output
uses LaTeX format.

**Core Assumption (must follow):**
This problem comes with a perfect geometric figure. **All point existence,
positional relationships, and topological structure (who intersects with
whom, who is on whom, who is collinear with whom) are clearly visible in the
figure.**
Therefore, any descriptions in the text used to **define point positions**
are redundant and **must be deleted**, even if this makes a point appear
"undefined" in the text.

1. **Delete (Visual/Topological Definitions)**:
   - Intersection definitions: e.g., "point G is the intersection of lines
     AB and CD", "intersect at point P" -> delete.
   - Position descriptions: e.g., "point D is on side BC", "A, B, C are
     collinear", "as shown in the figure", "in the plane" -> delete.
   - Construction processes: e.g., "draw segment AD", "connect BD" -> delete.

2. **Retain (Geometric Constraints/Metrics)**:
   - Metrics: numerical values (lengths, angles, areas), ratios.
   - Relationships: parallel (\parallel), perpendicular (\perp), equality (=).
   - Implicit metric high-level properties: must retain "midpoint" (implies
     1:1), "angle bisector" (implies equal angles), "tangent", "regular
     polygon", "diameter", etc.

3. **Rewrite (Natural Flow)**:
   - Use "given", "satisfy", "where", "and" to connect remaining conditions.
   - Keep sentences fluent, do not write as broken keyword lists.
   - **Maintain LaTeX format for all mathematical symbols**

**Example Learning (Few-Shot)**:

Example 1 (Delete intersection definition)
Input:
Given $AB=3$, $AC=6$, $\angle BAC=90^\circ$, point $D$ is the midpoint of
$AB$, point $G$ is the intersection of $CD$ and $AE$. Find the length of $GD$.
Output:
Given $AB=3$, $AC=6$, $\angle BAC=90^\circ$, and $D$ is the midpoint of $AB$.
Find the length of $GD$.

Example 2 (Delete construction and collinearity)
Input:
Given points $A$, $B$, $C$, $D$ in the plane, satisfying $AB \parallel CD$,
$AB = CD = 4$. Connect $AC$ and $BD$ intersecting at point $O$. If $A$, $O$,
$C$ are collinear, find the value of $\frac{AO}{OC}$.
Output:
Given $AB \parallel CD$, and $AB = CD = 4$. Find the value of $\frac{AO}{OC}$.

Example 3 (Retain circle and tangent)
Input:
As shown, $PA$, $PB$ are tangents to circle $O$, with points of tangency $A$,
$B$ respectively. Line $PO$ intersects circle $O$ at points $C$, $D$. If
$PA=4$, find the length of $PB$.
Output:
$PA$, $PB$ are tangents to circle $O$, given $PA=4$. Find the length of $PB$.

================= Problem Statement to Process (Original) =================
{question}

```
================= Output Format =================
Please output only a standard JSON object:
{
    "question_sanitized": "Rewritten concise problem statement (must maintain
                          LaTeX format)"
}
```

## VisualizeQA Prompt (Step 2: Annotation Filtering)

You are a professional mathematics editor, continuing to perform **Step 2 filtering** on the geometry problem. The filtered output **uses LaTeX format**.

**Important Notes:**
- You already have the simplified problem statement from Step 1 (denoted as Q1)
- Now you are given the **annotations** corresponding to this problem (image annotations, translated to natural language)
- annotations **only contain three types**: right angle annotations (perpendicular relationships), length annotations, angle annotations
- Your task is to **delete** all conditions from Q1 that are already explicitly annotated in annotations
- **When deleting conditions, you must maintain the LaTeX format of the remaining content unchanged**

**Core Rules:**
1. **Delete Principle**: If the problem statement explicitly mentions a condition that is in annotations (such as right angle, length, angle), and this condition has a corresponding annotation, then delete it from the problem statement
2. **Equivalence Recognition** (important!): Need to recognize equivalent expressions of geometric conditions. Even if the expression form is different, as long as there is an equivalent annotation in annotations, it should be deleted:
   - **Perpendicular and right angle are equivalent**:
     * `AD \perp CD` is equivalent to `\angle ADC = 90^\circ` or `\angle DAC = 90^\circ` or `\angle ACD = 90^\circ` (depending on which angle is the right angle)
     * `AB \perp BC` is equivalent to `\angle ABC = 90^\circ`
     * If annotations contain `\angle DAC = 90^\circ`, `AD \perp CD` in the problem statement should also be deleted
   - **Length equality**: `AB = CD` and `CD = AB` are equivalent
   - **Angle equality**: `\angle ABC = 45^\circ` and `\angle CBA = 45^\circ` are equivalent (same angle, different notation)
3. **Retain Principle**: If a description in the problem statement cannot be completely removed (e.g., the problem says "rectangle" but annotations only have one right angle annotation), then retain the description in the problem statement
4. **Numerical Retention**: If annotations only have annotation types (e.g., "right angle annotation: angleBAC = 90degrees"), but the problem statement has more specific numerical values or relationships (e.g., "AB=3"), as long as the numerical value is not in annotations, retain it

**Example Learning:**

Example 1 (Delete annotations)
Input Q1:
Given $AB=3$, $AC=6$, $\angle BAC=90^\circ$, and $D$ is the midpoint of $AB$. Find the length of $GD$.
annotations:
Right angle annotation: angleBAC = 90degrees; Length annotations: AB = 3, AC = 6
Output:
Given $D$ is the midpoint of $AB$. Find the length of $GD$.
(Deleted $\angle BAC=90^\circ$, $AB=3$ and $AC=6$, because they are all in annotations)

Example 2 (Equivalence recognition: perpendicular and right angle)
Input Q1:
Given $AD \perp CD$, and $AD=5$, $CD=12$. Find the length of $AC$.
annotations:
Right angle annotation: angleDAC = 90degrees
Output:

```
Given $AD=5$, $CD=12$. Find the length of $AC$.
(Deleted $AD \perp CD$, because $\angle DAC = 90^\circ$ is equivalent to
$AD \perp CD$, and annotations contain angleDAC = 90degrees)

Example 3 (Equivalence recognition: angle different notation)
Input Q1:
Given $\angle ABC = 45^\circ$, and $AB=5$. Find the length of $BC$.
annotations:
Angle annotation: angleCBA = 45degrees
Output:
Given $AB=5$. Find the length of $BC$.
(Deleted $\angle ABC = 45^\circ$, because $\angle ABC$ and $\angle CBA$ are
the same angle with different notation, equivalent)

Example 4 (Retain descriptions that cannot be completely removed)
Input Q1:
Given rectangle $ABCD$, $AB=5$, $BC=12$. Find the length of diagonal $AC$.
annotations:
Right angle annotation: angleABC = 90degrees
Output:
Given rectangle $ABCD$, $AB=5$, $BC=12$. Find the length of diagonal $AC$.
(Retain "rectangle", because the problem says rectangle, but annotations only
have one right angle annotation, cannot completely remove the rectangle concept)

================= Input: Simplified Problem Statement Q1 from Step 1 =================
{question_simplified}

================= Input: Corresponding annotations (image annotations, translated
to natural language) =================
{annotations}

================= Output Format =================
Please output only a standard JSON object:
{
    "question_sanitized": "Based on Q1, problem statement after deleting
                           conditions already annotated in annotations (must
                           maintain LaTeX format)"
}
```

## VisualizeQA Prompt (CoT Rewriting)

```
You are a **professional geometry VQA rewriting expert**.

Your task is:
Based on the input **plotting code (geometric construction code that generates
the image)**, strictly rewrite the original **CoT** to generate a reasoning chain
suitable for visual question answering format.
=====================================================
Input Data
=====================================================
Original reasoning process (cot):
{cot}
Plotting code (geometric construction code that generates the image, represents
the geometric structure and relationships visible in the image):
{plotting_code}
=====================================================
[Rewriting Rules: CoT] (Visual-integrated Reasoning)
=====================================================
Based on original CoT + plotting code, rewrite the reasoning process:
1. **Allow directly using "geometric facts" from plotting code as known
   conditions, starting with "From the image..." or "From the plotting code..."**
   Used to simplify the reasoning path.
2. The reasoning process must use standard format:
   `Step 1: ...`
   `Step 2: ...`
   Each step only performs one logical action.
3. **Do not infer geometric information not present in plotting code**
   Structures, relationships, or properties not defined in plotting code must not
   be treated as visual facts.
4. Only use the following sources for reasoning:
   - Geometric structures and relationships defined in plotting code
   - Implicit conditions from the problem statement
   - General Euclidean geometry knowledge (such as triangle angle sum,
     similarity definition, etc.)
5. **Delete some redundant reasoning steps, keep compact but correct.**
6. The final answer must match the original problem.

=====================================================
[Output Format](Must strictly follow)

Please output standard JSON:
{
"cot": "Step 1: ...\nStep 2: ..."
}
```

## VL Image Quality Prompt (Quality Check)

```
You are a professional geometry image quality assessment expert. Please judge
based on the visual content of the image itself whether this image is suitable
as an illustration for a geometry problem.

=====================================================
Image Quality Assessment Requirements
=====================================================

Please strictly judge based on "the visual content of the image itself"
whether it passes the quality check.

Assessment dimensions:
- Are lines clear, unbroken, and not blurred?
- Are point names, line segments, annotations, etc., severely occluded?
- Are annotations readable?
- Is the figure layout crowded or chaotic?

Judgment rules:
- If any "severely affects understanding" issue appears, output `"passed": false`
- Otherwise output `"passed": true`
- Briefly explain the reason, do not perform reasoning

=====================================================
Output Format (must strictly follow, output only JSON)
=====================================================

{
  "passed": true/false
  "reason": "Brief explanation of reason"
}
```

## VL Image Quality Prompt (Caption Generation)

```
You are a professional geometry image description expert. Please generate an
objective, complete, non-inferential natural language description based on
visible image content and plotting_code.

==================================================
Plotting Code (Structure Reference)
==================================================
{plotting_code_str}

==================================================
Task Requirements
==================================================

Please generate an objective, complete, non-inferential natural language
description based on visible image content.

Requirements:
- Only describe points, lines, circles, annotations actually visible in the
  image
- Do not reference elements in plotting_code that are not actually visible
  in the image
- Do not infer relationships like equality, parallelism, perpendicularity
  if not explicitly shown
- Do not complete geometric information not present in the figure
- Do not use problem context, do not create narrative explanations
- Avoid hallucinations, strictly based on visible image content

==================================================
Output Format (must strictly follow, output only JSON)
==================================================

{
  "caption": "Image description"
}
```

