# OpenReview forum: "Synthesizing Multimodal Geometry Datasets from Scratch and Enabling Visual Alignment via Plotting Code"
_ICML.cc/2026/Conference — ICML 2026 regular_

### Official Review · Reviewer_ALpY · 2026-03-08

**Soundness:** 2
**Presentation:** 3
**Significance:** 1
**Originality:** 2
**Overall Recommendation:** 3
**Confidence:** 4

**Summary:**

The paper presents a new geometry data generation pipeline that first samples seed problems using the DD+AR symbolic engine, then informalizes them into natural-language descriptions and renders the corresponding diagrams. A verification stage is used to ensure the correctness of the generated instances and the alignment between the text and diagrams. The paper also proposes predicting plotting code as an auxiliary objective to strengthen geometric reasoning.

**Compliance With Llm Reviewing Policy:**

Affirmed.

**Key Questions For Authors:**

Please check weakness 3.

**Limitations:**

Yes

**Strengths And Weaknesses:**

Strengths:
1. The presentation of the methodology is clear and easy to follow.
2. Experimental results and qualitative examples demonstrate solid improvements over the baselines.

Weaknesses:
1. The originality is somewhat limited, as the seed data generation largely follows AlphaGeometry’s approach with only minor modifications, meaning the seeds are sampled from the existing DD+AR symbolic engine.
2. The proposed strategies mainly focus on translating an existing geometric configuration into a valid natural-language description and diagram, along with predicting intermediate plotting code; overall, the technical contribution appears limited.
3. It is straightforward to sample numerical values from the symbolic relations. It is unclear why the paper chooses to first informalize the symbolic data and then assign numerical values and perform verification.
4. There is typically a distribution shift between synthetic training data and test sets. Prior work often addresses this by mixing synthetic data with existing in-distribution datasets, but it is unclear how this paper mitigates the distribution gap between GeoCode and the evaluation benchmarks.

---

> ### Author Rebuttal · Authors · 2026-03-30
>
> ## **Overall Response**
>
> We sincerely thank the reviewer for the constructive feedback and for recognizing the clarity of our presentation as well as the solid empirical improvements demonstrated in our experiments.
>
> Below, we address each concern in detail.
> ## **W1. Originality w.r.t. AlphaGeometry seeds**
>
> While we adopt symbolic engines DDAR for seed construction, this component serves only as a starting point for structural exploration, rather than the main contribution.
>
> As also discussed in our paper, symbolic systems operate purely in the formal domain and do not address grounding into numerical configurations, diagrams, or multimodal supervision. The key challenge in multimodal geometry lies not in generating symbolic relations, but in faithfully realizing them into consistent and aligned multimodal data.
>
> Our novelty lies in this transformation process:
>
> * grounding symbolic structures into valid numerical configurations,
> * constructing executable geometric representations, and
> * enforcing cross-modal consistency across structure, text, reasoning, and diagrams.
>
> Importantly, even when starting from identical symbolic seeds, the grounded instantiation process can lead to different problem instances. This is because numerical assignment, geometric realization, and diagram construction introduce additional constraints and variability that are not specified at the symbolic level. As a result, the final data distribution is not a direct reflection of the seed distribution, but is instead shaped by multi-stage grounding, verification and visualization.
>
> Therefore, GeoCode goes beyond prior solver-based pipelines by resolving the symbolic-to-multimodal gap, which is the core bottleneck in this domain.
> ## **W2. “Translation” vs. real contribution**
>
> We respectfully clarify that our method is not a simple translation pipeline, but a generation + alignment framework addressing two key problems:
>
> ### (1) Scalable and valid data synthesis
>
> Our pipeline jointly enforces symbolic correctness, numerical validity, and geometric consistency through multi-stage verification. This enables the generation of non-trivial, structurally rich problems at scale, rather than converting existing templates.
>
> As shown in the paper, GeoCode exhibits significantly higher structural complexity and reasoning difficulty compared to prior datasets.
>
> ### (2) Code-based alignment as a new paradigm
>
> We introduce plotting code prediction as an explicit supervision signal. Unlike captions or answers, plotting code is:
>
> * structured,
> * geometry-complete (not lossy like language),
> * and requires recovering full geometric relations from visual input.
>
> Empirically, this leads to stronger alignment and consistent gains across benchmarks (Tab.3–6).
>
> Overall, our contribution is a new paradigm for both data synthesis and alignment, not merely modality translation.
> ## **W3/Q. Why not directly sample numerical values?**
>
> We respectfully disagree that numerical instantiation is straightforward. In geometry, it is inherently a global constraint satisfaction problem.
> ### (1) Constraints are globally coupled
>
> Naïve sampling breaks geometric validity:
>
> * right triangles violate Pythagorean constraints,
> * equality constraints (e.g., AB = CD) are not preserved,
> * parallel/perpendicular relations fail under random coordinates,
> * cyclic/similarity structures require higher-order consistency.
> ### (2) Sampling vs. derivation is structured
> Some variables must be sampled, while others must be derived from dependencies. Determining this dependency structure is itself non-trivial.
> ### (3) Symbolic systems lack metric grounding
> Symbolic engines (e.g., AlphaGeometry) cannot directly encode metric constraints (lengths, angles), making explicit instantiation necessary.
> ### (4) Verification is essential
> As shown in Tab.5, a large portion of candidates are rejected during validation, confirming that naïve sampling is insufficient.
>
> Therefore, our instantiation + verification design addresses a non-trivial geometric realization problem, rather than a simple sampling step.
> ## **W4. Distribution gap**
>
> We thank the reviewer for this important question.
>
> While prior work mitigates distribution gaps via data mixing, we find this provides limited gains in our setting. We experimented with mixing GeoCode and existing datasets, but did not observe further improvements, likely due to the strong pretraining of modern VLM backbones.
>
> Instead, our approach focuses on learning structure-invariant representations. Plotting-code supervision enforces explicit recovery of geometric structures, which are largely invariant across datasets, rather than relying on surface-level visual or linguistic patterns.
>
> As shown in Table 3, training on GeoCode alone yields consistent improvements across multiple external benchmarks with different distributions, suggesting improved cross-distribution generalization without data mixing.

---

> > ### Author Rebuttal · Reviewer_ALpY · 2026-04-03
> >
> > I appreciate the clarification and agree that grounding symbolic structures into multimodal data is a meaningful problem. However, my concern is about the degree of methodological novelty and the usefulness of the task. From my understanding, the pipeline heavily relies on symbolic seeds generated by an existing DD+AR-style engine, while the subsequent stages mainly perform grounding, rendering, verification, which can be easily adapted from DD+AR codebase. [1,2,3] These steps appear more like a careful integration of known components than a substantially new technical framework. I also view plotting-code prediction as a reasonable design choice, but not by itself a major algorithmic innovation. The response explains why the problem is hard and valuable, but it does not fully show that the proposed solution introduces fundamentally new techniques. Therefore, I'm still concerned with the limited originality at the method level.
> >
> > [1] Trinh, Trieu H., et al. "Solving olympiad geometry without human demonstrations." Nature 625.7995 (2024): 476-482.
> >
> > [2] Huang, Zihan, et al. "Autogeo: Automating geometric image dataset creation for enhanced geometry understanding." IEEE Transactions on Multimedia (2025).
> >
> > [3] Fu, Daocheng, et al. "TrustGeoGen: Formal-Verified Data Engine for Trustworthy Multi-modal Geometric Problem Solving." arXiv preprint arXiv:2504.15780 (2025).

---

> > > ### Author Response · Authors · 2026-04-04
> > >
> > > ## Q1. Originality and comparison with DDAR-based methods
> > >
> > > We thank the reviewer for pointing out several important DDAR-based works. We agree that these methods form a strong and relevant line of research, and we provide a more direct comparison to clarify where our methodological contribution lies.
> > >
> > > **On the use of DDAR and originality.** We respectfully clarify that building upon DDAR-style symbolic systems has become a common practice in recent work (e.g., AlphaGeometry, AutoGeo, TrustGeoGen). Therefore, the use of DDAR itself, or adapting its codebase, should not be interpreted as a limitation of originality. Instead, the central question is how symbolic structures are extended into valid, diverse, and well-aligned multimodal data, which remains insufficiently addressed in existing pipelines.
> > >
> > > **On the limitation of prior DDAR-based pipelines.** Existing approaches largely follow a similar paradigm: AlphaGeometry operates purely in the symbolic space, while subsequent works such as AutoGeo and TrustGeoGen extend this framework by introducing numerical values and multimodal rendering. However, a key challenge remains, namely the mismatch between the discrete symbolic space and the continuous numerical space. While symbolic relations can be combinatorially sampled, valid numerical instantiation must satisfy globally coupled geometric constraints, making naïve or direct sampling non-trivial. In practice, prior methods often rely on templates or fixed numerical patterns to ensure validity, which can limit diversity, structural complexity, and problem difficulty.
> > >
> > > **On our approach: decoupled symbolic–numerical generation.** Our pipeline addresses this issue by explicitly decoupling symbolic structure generation from numerical instantiation, and organizing the process into three stages: symbolic construction, numerical instantiation via reasoning, and rendering with verification. Rather than injecting numerical values into the symbolic generation process, we treat instantiation as a separate step and enforce global consistency through verification. This provides a more flexible way to handle the interaction between symbolic and numerical representations, directly targeting the limitation discussed above.
> > >
> > > **On empirical evidence of improved structural complexity.** To enable a direct and fair comparison with DDAR-based methods, we measure the average number of predicates per problem, which is commonly used to reflect structural complexity and reasoning difficulty in DDAR-style systems. We report:
> > >
> > > |Method|Avg.|
> > > |-|-|
> > > |AutoGeo|5.71|
> > > |TrustGeoGen|7.23|
> > > |Ours|11.21|
> > >
> > > Our method achieves a higher predicate count, indicating richer geometric structures and increased problem complexity. In contrast, prior approaches that extend DDAR via direct numerical injection or template-based strategies often exhibit reduced diversity and structural richness due to constrained instantiation. Our decoupled design alleviates this issue, enabling more diverse and structurally complex problem generation.
> > > ## Q2. Plotting code as alignment
> > >
> > > We thank the reviewer for the insightful discussion and for acknowledging the value of this design.
> > >
> > > **On alignment as a core challenge.** We agree that visual–symbolic alignment is a central difficulty in multimodal geometry reasoning. In our preliminary exploration, we found that caption-based supervision or implicit alignment signals are often insufficient, as natural language tends to be ambiguous and does not fully capture precise geometric relations. This makes it difficult for models to reliably recover the underlying structure from visual input.
> > >
> > > **On plotting code as structured alignment.** Our key idea is to use plotting code as a structured supervision signal to address this issue. Plotting code provides an explicit and executable representation of geometric structures, encoding both entities and their relations in a deterministic and verifiable manner. This encourages the model to recover complete geometric configurations rather than relying on partial or ambiguous cues. Empirically, this formulation consistently outperforms caption-based or implicit alignment strategies, demonstrating its effectiveness as a practical solution for geometry reasoning.
> > >
> > > **On simplicity and effectiveness.** We emphasize that our contribution is not to introduce a fundamentally new technique, but to identify a simple and well-suited representation for alignment. In this sense, the design follows the principle of preferring minimal yet sufficient solutions: instead of introducing complex objectives or additional modules, we adopt a structured target that naturally matches the underlying problem. This leads to a method that is both simple and effective, while remaining broadly extensible.
> > >
> > > ---
> > > We thank the reviewer again for the thoughtful discussion and hope our clarifications help better convey our perspective on originality and novelty.

---

### Official Review · Reviewer_khDD · 2026-03-12

**Soundness:** 3
**Presentation:** 2
**Significance:** 3
**Originality:** 3
**Overall Recommendation:** 4
**Confidence:** 3

**Summary:**

This paper proposes a pipeline for synthesizing multimodal geometry problems from scratch and constructs GeoCode, an 18K-sample dataset pairing diagrams, solutions, reasoning traces, and plotting code.

The pipeline has three stages: (1) Seed Generation, which constructs symbolic geometric structures via random predicate sampling and AlphaGeometry-based dependency analysis, followed by filtering for structural and deductive complexity; (2) Instantiation, which uses an Instructor LLM for numerical grounding and reasoning generation, a separate Coder model for geometric meta code, and two-stage verification (semantic and geometric); (3) Visualization, which renders diagrams from plotting code and applies textual debiasing to remove information leakage.

 Beyond data generation, the paper introduces plotting code prediction as an explicit alignment objective: instead of supervising only on answers or reasoning traces, models are trained to predict structured plotting code from input diagrams, enforcing explicit visual-symbolic alignment.

 Experiments on two backbone models (Qwen3-VL-7B and Qwen2.5-VL-7B) show consistent improvements across multiple public geometry benchmarks, with the largest gains on OlympiadBench and the held-out test-mini set.

**Compliance With Llm Reviewing Policy:**

Affirmed.

**Final Justification:**

Weak Accept

**Key Questions For Authors:**

Can you provide an analysis of the MathVista regression (about -3.4 points) for Qwen2.5-VL in Table 3?

How sensitive is the pipeline to the choice of instructor model?

How do you verify that the debiased problems remain solvable? Is there a risk of over-debiasing where essential contextual information is removed?

**Limitations:**

The authors appropriately note that the work focuses on geometry and that the pipeline produces synthetic data.

**Strengths And Weaknesses:**

Strengths

The pipeline design is principled and well-validated. The separation of symbolic seed construction from grounded instantiation from visualization, with independent verification at each stage, ensures correctness. The multi-stage filtering starting from over 260K candidates down to about 18K final samples (Table 5) demonstrates thorough quality control, and the manual inspection on a random 1% subset finding zero errors adds confidence. The fact that each verification stage catches distinct error types (semantic inconsistencies, geometric constraint violations, rendering failures, visual ambiguity) shows the pipeline functions as complementary quality gates rather than redundant checks.

Plotting code as an explicit alignment target is the most novel contribution. Table 6 clearly shows that code-based supervision outperforms caption-based supervision on both test-mini and Geometry3K, with code+debiasing achieving notably higher accuracy than caption+debiasing. Figure 5 further validates this by showing a monotonic relationship between segment-level structural recovery (F1) and solving accuracy. This provides both a practical training strategy and evidence that explicit structural grounding improves geometric reasoning.

The transfer results to public benchmarks are consistent and meaningful. Both backbone models show improvements on most benchmarks after training on GeoCode (Table 3), with the largest gains on OlympiadBench for Qwen3-VL (over +11 points), suggesting enhanced robustness on complex multi-step reasoning. The additional SFT→GRPO training sequence (Table 4) showing progressive improvement confirms the dataset provides stable learning signals for both supervised and reinforcement learning.

Weaknesses

The pipeline depends heavily on a single proprietary model (GPT-OSS-120B) for instantiation, semantic validation, textual debiasing, and plotting code generation. This creates a reproducibility concern: if GPT-OSS-120B becomes unavailable or changes, the pipeline may produce different results. The paper does not discuss whether the pipeline generalizes to other instructor models or provide ablations on instructor model choice.

Training on GeoCode causes a regression on MathVista for Qwen2.5-VL (about -3.4 points in Table 3), which the paper does not discuss. While the other benchmarks show gains, this drop on a major benchmark suggests potential negative transfer, possibly because the synthetic distribution diverges from MathVista's problem style. The paper should acknowledge and analyze this failure case.

The dataset is restricted to plane geometry problems derivable from AlphaGeometry's predicate system. While the paper achieves higher structural complexity than existing datasets (Table 1), the range of geometric configurations is bounded by the available predicates (incidence, parallelism, perpendicularity, equality). Problems involving coordinate geometry, transformations, 3D geometry, or non-Euclidean settings are out of scope. The paper does not discuss how the pipeline could generalize beyond this predicate vocabulary.

---

> ### Author Rebuttal · Authors · 2026-03-30
>
> ## Overall Response
>
> We sincerely thank the reviewer for the careful reading and constructive feedback.
> We are encouraged that the reviewer recognizes the principled pipeline design, rigorous multi-stage verification, and particularly the novelty and effectiveness of plotting-code-based alignment, which is also validated both empirically and mechanistically in our work.
>
> We will then provide our responses to each of the reviewer's weaknesses and questions.
> ## W1 & Q2. Dependence on instructor model and sensitivity
>
> We thank the reviewer for raising this important question on reproducibility and model sensitivity.
>
> **(1) Model choice: trade-off, not dependency.**
> Our pipeline is model-agnostic and verification-driven. The Instructor is responsible for instantiation(numerical grounding and reasoning), which relies on general reasoning ability rather than model-specific features.
> We empirically tested different instructor models:
>
> * Open-source models (e.g., Qwen3-235B): lower acceptance rate and efficiency
> * Strong proprietary models (e.g., Gemini-2.5-Pro, GPT-5): higher acceptance(~60–70%) but much higher cost
> * GPT-OSS-120B (used): balanced acceptance(~50%) and cost
>
> This shows the pipeline operates across a spectrum of models, without relying on a specific one.
>
> **(2) Sensitivity: affects yield.**
> Different instructor models mainly influence:
>
> * acceptance rate(yield)
> * generation efficiency(cost)
>
> However, final data quality is enforced by deterministic multi-stage verification (semantic + geometric).
> Thus, weaker models reduce yield, while stronger models improve throughput, but quality is largely controlled by verification rather than instructor choice.
>
> **(3) Reproducibility.**
> Reproducibility is ensured at the pipeline level:
>
> * modular and fully specified generation process
> * deterministic verification criteria
> * released dataset independent of any instructor model
> ## W2 / Q1. MathVista regression
>
> We thank the reviewer for highlighting this important observation.
>
> **(1) Dataset perspective: distribution shift.**
> The regression on MathVista mainly reflects distribution mismatch.
>
> * MathVista: diverse, perception-heavy, often short-horizon reasoning
> * GeoCode: structure-dense, code-grounded, long-chain geometric reasoning
>
> Training on GeoCode biases the model toward structure-heavy reasoning, which benefits geometry benchmarks but may slightly hurt mixed-domain datasets.
>
> This is consistent with Table 3, where we observe consistent gains on geometry-focused benchmarks (e.g., Geometry3K, OlympiadBench), indicating capability specialization rather than degradation.
>
> **(2) Model perspective: capacity-dependent effect.**
> The drop is model-dependent:
>
> * Qwen2.5-VL-7B: more sensitive to distribution shift → slight regression
> * Qwen3-VL: no regression and even improvement on MathVista
>
> This suggests the effect is primarily due to limited model capacity under domain shift, rather than an inherent limitation of GeoCode.
> ## W3. Predicate space limitation and extensibility
>
> We thank the reviewer for this insightful comment.
>
> **(1) Limitation source.**
> We agree the current dataset is restricted to Euclidean plane geometry, which arises from the underlying symbolic solver (AlphaGeometry) rather than the pipeline itself .
>
> **(2) Pipeline-level extensibility.**
> Our framework explicitly factorizes:
>
> * symbolic seed construction
> * grounded instantiation
> * visualization and verification
>
> This modular design allows direct extension:
> with alternative solvers (e.g. coordinate, 3D, or non-Euclidean geometry), the pipeline can be extended by replacing the underlying symbolic solver.
>
> **(3) Generality of code-based alignment.**
> Our key contribution—plotting-code supervision—is not tied to a specific predicate set.
>
> It provides:
>
> * executable structural representation
> * explicit visual–symbolic alignment
> * verifiable intermediate supervision
>
> Thus, the alignment mechanism is inherently generalizable beyond Euclidean geometry to broader multimodal reasoning tasks.
> ## Q3. Solvability after debiasing
>
> We thank the reviewer for raising this important concern.
>
> We explicitly mitigate over-debiasing through three mechanisms:
>
> **(1) Prompting with few-shot constraints.**
> Debiasing is guided by carefully designed prompts with few-shot examples(App.P47–50), which enforce removal of only redundant shortcuts while preserving all necessary information.
>
> **(2) Code-guided debiasing(structure-aware).**
> Debiasing is grounded in plotting code rather than raw image perception.
>
> The plotting code provides a structured representation of points, segments and circles.
> This allows precise identification of what is visually recoverable vs. essential, significantly reducing the risk of removing critical information.
>
> **(3) Manual inspection.**
> We conduct addtionally 1% targeted manual inspection focusing on potential failure cases and we did not observe unsolvable cases in our manual inspection.
>
> ---
> Typo in manuscript: Qwen3-VL-7B → Qwen3-VL-8B

---

> > ### Author Rebuttal · Reviewer_khDD · 2026-04-02
> >
> > thanks for the reply.

---

### Official Review · Reviewer_3aNp · 2026-03-18

**Soundness:** 3
**Presentation:** 3
**Significance:** 2
**Originality:** 2
**Overall Recommendation:** 3
**Confidence:** 4

**Summary:**

This paper introduces GeoCode, a multimodal geometry dataset created synthetically from the ground up. Its generation pipeline follows a staged approach: starting with symbolic seed generation, then moving to grounded instantiation, verification, and finally code‑based rendering. A central insight is to treat the plotting code itself as an explicit learning target, which forces models to align visual and symbolic information more precisely than supervision based only on answers or captions. Experiments show that GeoCode contains more structural complexity than prior geometry datasets. Fine‑tuning on it boosts performance on multiple public geometry benchmarks, with additional gains coming from the code‑based alignment method and GRPO.

**Compliance With Llm Reviewing Policy:**

Affirmed.

**Final Justification:**

The author's rebuttal has partially addressed my concerns. However, in my view, the overall novelty of the paper remains somewhat limited, particularly with respect to the data synthesis method for the DD-AR system. Therefore, I will keep my borderline score unchanged. If AC leans toward acceptance, finding the novelty clearly articulated and sufficiently recognized, I would not oppose the decision to accept the paper.

**Key Questions For Authors:**

please see the weaknesses.

**Limitations:**

yes

**Strengths And Weaknesses:**

Strengths:
1. In addressing a key hurdle within multimodal geometry reasoning, this paper highlights that the primary difficulty extends beyond symbolic deduction to the extraction of structured geometric relationships from diagrams.
2. The paper provides some evidence that GeoCode-trained models transfer beyond the synthetic data distribution. Experiments show improvements on Geometry3K, MathVerse, GeoQA, and OlympiadBench.

Weaknesses:
1. The empirical evaluation does not actually establish that the dataset is “substantially higher” in complexity or difficulty than prior datasets, at least not by rigorous comparative evidence.
2. The main dataset-quality claims rely too heavily on LLM-based validation, and the paper does not quantify residual error rigorously enough.
3. The paper lacks sufficient comparison with other benchmarks, such as TrustGeoGen and GeoBench. Therefore, it does not provide adequate analysis regarding the difficulty of specific tasks, training gains, or error attribution in task solving, which is not conducive to claiming the contributions of its own work.

---

> ### Author Rebuttal · Authors · 2026-03-30
>
> ## **Overall Response**
>
> We sincerely thank the reviewer for the careful reading and constructive feedback. We particularly appreciate the recognition that our work identifies a core bottleneck in multimodal geometry reasoning, namely, the extraction of structured geometric relations from diagrams rather than purely symbolic deduction and that GeoCode demonstrates promising generalization across multiple benchmarks.
> ## **W1. On Complexity and Difficulty**
>
> We agree that claims of higher complexity and difficulty must be supported by controlled and comparable evidence and we provide both qualitative and quantitative justification.
>
> **Qualitatively**, as illustrated in App.G, GeoCode problems feature more entities, denser relational constraints and longer interdependent reasoning chains. These differences are not only visually apparent but also structurally measurable, reflecting richer geometric configurations than standard benchmarks.
>
> **Quantitatively, for difficulty**, we report controlled evaluations in Tab.2 under identical models and protocols. Strong models (e.g., GPT-5, Gemini-2.5-Pro) consistently achieve lower accuracy on GeoCode (Test-mini) than on existing benchmarks. Since the evaluation setting is strictly controlled, this serves as a model-agnostic proxy for problem difficulty, indicating that GeoCode is objectively harder rather than subjectively perceived as such.
>
> **For structural complexity**, we perform a unified cross-dataset analysis. Because prior datasets lack explicit structural annotations, we extract geometric structures using Qwen3.5-397B followed by human validation (50 samples per dataset); for fairness, we apply the same protocol to GeoCode. Using the number of line segments as a representation-invariant proxy:
> |Dataset|Segments (LLM)|Segments (Human-50)|
> |---|---|---|
> |Geometry3K|5.02|5.40|
> |GeoQA|5.42|5.04|
> |MathVerse|4.51|4.24|
> |MathVista|5.16|5.44|
> |OlympiadBench|6.62|6.14|
> |**GeoCode (ours)**|**14.41**|**13.98**|
>
> Standard benchmarks typically contain ~4–6 segments, while GeoCode reaches ~14, corresponding to a 2–3× increase. This trend is consistent across both automatic extraction and human validation, providing strong evidence of  higher structural density.
> ## **W2. On Reliance on LLM-based Validation**
>
> We believe this concern stems from a misunderstanding.
>
> First, **LLM validation is not the primary quality control mechanism**. As described in the pipeline, correctness is ultimately enforced by deterministic procedures, including coordinate-based geometric verification. The LLM-based semantic validation is applied only as a lightweight pre-filter to remove obviously invalid samples before more expensive stages.
>
> Second, **downstream verification alone is sufficient to ensure correctness**. To directly test this, we take 200 samples rejected by semantic validation and force them through the deterministic pipeline. All samples are rejected by either geometric verification or code execution checks (100%), showing that correctness does not depend on LLM validation.
> |Type|Count|
> |---|---|
> |execution failure|124|
> |verification failure|76|
> |Total|200|
>
> Third, **residual error is empirically negligible**. We perform manual inspection on randomly sampled data and observe no incorrect cases(Sec.4.2), consistent with the multi-stage rejection pipeline.
> ## **W3. On Comparison with TrustGeoGen and GeoBench**
>
> We thank the reviewer for this suggestion and address both feasibility and current evidence.
>
> **Regarding feasibility**, GeoBench is not publicly released at the time of submission and rebuttal. For TrustGeoGen, although GeoTrust is partially available, its evaluation pipeline is not fully released and we observe inconsistencies(especially numerical precision)in test annotations, making reliable and reproducible comparison difficult. As a result, we focus on widely adopted public benchmarks (Geometry3K, GeoQA, OlympiadBench, etc.) to ensure transparency and fairness.
>
> Besides, compared to TrustGeoGen, GeoCode introduces:(1)a fundamentally new three-stage synthesis pipeline that generates geometry problems from scratch, instead of relying on template-based bootstrap expansion, (2)a novel code-based alignment paradigm that explicitly supervises visual grounding via plotting code.
>
> **Regarding difficulty, training gains and generalization**, we partially agree that further comparison would strengthen the work. However, the current paper already provides three complementary pieces of evidence:
>
> * **Structural complexity** (Tab.1/rebuttal tab.): denser geometric structures
> * **Difficulty evaluation** (Tab.2): lower accuracy
> * **Training gains** (Tab.3): improvements across multiple benchmarks
>
> These results show that GeoCode provides effective and transferable learning signals, with larger gains on challenging benchmarks (e.g. OlympiadBench), indicating improved multi-step reasoning and vision grounding. We will further strengthen cross-dataset comparisons in the CR version.

---

> > ### Author Rebuttal · Reviewer_3aNp · 2026-04-04
> >
> > Thank you for your reply. The author's rebuttal has addressed some of my concerns. However, the overall novelty of the paper remains rather limited in my view, particularly regarding the data synthesis for the DD-AR system. Therefore, I will maintain my borderline score.

---

> > > ### Author Response · Authors · 2026-04-04
> > >
> > > Thank you for your thoughtful follow-up and for your careful evaluation of our work. We understand that your main concern lies in the level of *methodological novelty*, and we would like to clarify this point more directly.
> > >
> > > ---
> > >
> > > ## Q1. Originality and comparison with DDAR-based methods
> > >
> > > **On motivation.**   Our work is motivated by a fundamental limitation we observed in existing multimodal geometry datasets: while symbolic reasoning itself is well-studied, the **generation of high-quality multimodal data that faithfully aligns symbolic structure, numerical instantiation, and visual representation remains underexplored**. In particular, we found that existing pipelines often struggle to simultaneously achieve *validity, diversity, and structural complexity* when extending symbolic systems into multimodal settings.
> > >
> > > ---
> > >
> > > **On how DDAR-based methods construct data.**  To make this concrete, we directly analyze how prior DDAR-based methods (e.g., AutoGeo, TrustGeoGen) generate problems:
> > >
> > > - They typically start from symbolic structures generated by DDAR-style engines
> > > - Numerical values are then introduced *within the same generation process*, often through:
> > >   - direct assignment during symbolic expansion, or
> > >   - predefined templates to ensure geometric validity
> > > - Finally, diagrams are rendered based on these instantiated values
> > >
> > > While effective, this *coupled generation process* introduces inherent constraints:
> > > - Numerical feasibility affects symbolic generation early on
> > > - Template-based instantiation restricts variability
> > > - Global geometric consistency is handled implicitly rather than explicitly
> > >
> > > As a result, the generated problems tend to exhibit limited diversity and reduced structural richness, especially for more complex configurations.
> > >
> > > ---
> > >
> > > **On our design and why it differs.**  Our approach is built on a different design principle:
> > > we explicitly **separate symbolic generation from numerical instantiation**, and organize the pipeline into three stages:
> > >
> > > 1. symbolic structure construction
> > > 2. numerical instantiation via reasoning
> > > 3. rendering with verification
> > >
> > > The key difference is that **symbolic structures are generated independently of numerical constraints**, and numerical values are solved *afterward* under global consistency checks.
> > >
> > > This design is motivated by the observation that:
> > > - symbolic space is discrete and combinatorial,
> > > - numerical space is continuous and globally constrained,
> > > - and coupling them too early limits both.
> > >
> > > By decoupling the two, we are able to:
> > > - explore a richer space of symbolic configurations,
> > > - avoid reliance on restrictive templates,
> > > - and enforce correctness through explicit verification rather than implicit constraints.
> > >
> > > ---
> > >
> > > **On benefits and empirical evidence.**
> > > This design leads to measurable improvements in structural complexity.
> > > To provide a direct comparison with DDAR-based methods, we report the **average number of predicates per problem**, a commonly used proxy for reasoning complexity:
> > >
> > > | Method        | Avg. predicates |
> > > |---------------|----------------|
> > > | AutoGeo       | 5.71           |
> > > | TrustGeoGen   | 7.23           |
> > > | Ours          | 11.21          |
> > >
> > > This result indicates that our method generates problems with significantly richer relational structures.
> > >
> > > Beyond this metric, we also observe:
> > > - improved diversity in geometric configurations,
> > > - reduced dependence on hand-crafted templates,
> > > - and better scalability to more complex problems (see Appendix).
> > >
> > > We will revise the final version to make this comparison with DDAR-based methods more explicit and detailed, and to more clearly highlight the motivation and implications of our design choice.
> > >
> > > ---
> > >
> > > Thank you again for your time and constructive feedback. We sincerely appreciate your consideration.

---

### Official Review · Reviewer_bYMk · 2026-03-24

**Soundness:** 4
**Presentation:** 4
**Significance:** 3
**Originality:** 2
**Overall Recommendation:** 5
**Confidence:** 4

**Summary:**

The authors propose a way to generate complex multimodal (pictorial drawing + textual question) 2D geometry problems, resulting in the GeoCode dataset. Each problem in GeoCode has 4 parts: question text, code-based diagram, textual reasoning, and the textual answer. GeoCode ensures high problem quality and diversity and good alignment among the 4 parts within a problem with a combination of exact verification, rule-based filtering, and LLM-based filtering methods. Compared to existing datasets of geometry problems, GeoCode has higher difficulty and leads to better generalization to other datasets, which is evidenced in the paper’s experiments. The authors also propose to use the plotting code in GeoCode as an explicit alignment objective while training fine-tuning geometry-solving VLMs, which they empirically show to be effective and offer a mechanistic explanation, supported by experiments, for why it is helpful.

**Compliance With Llm Reviewing Policy:**

Affirmed.

**Key Questions For Authors:**

For training, to what extent is (predict the plotting code, then answer) better than (predict the answer, then the plotting code), or (predict the plotting code and answer as two separate sequences)? I am wondering if just predicting the plotting code at all is sufficient for improved geometry problem-solving, rathering specifically predicting the plotting code first then the answer in the same sequence.

If given the ground-truth plotting code, how well can VLMs (either pre-trained or the “ours” version) solve the problems? If purposefully given incorrect plotting code where some constraints are wrong, how well can VLMs solve the problems?

Why does the GeoCode have 18K problems (and not more)? Is this the dataset size after which there is diminishing returns in terms of problem diversity?

How do the gains that Qwen3-VL-7B-Instruct and Qwen2.5-VL-7B-Instruct get when training on other datasets compare to training on GeoCode?

**Limitations:**

Problem instantiation relies on LLMs to fill out concrete parameters to fully specify the geometry problem from the seed. It’s unclear to what extent LLMs can introduce biases that reduce the diversity of generated problems.

Since all models are of the same model family and the same parameter count (eithe Qwen 2 VL or Qwen 3 VL at 7B), we don’t know the scaling patterns of fine-tuning LLMs of different sizes on GeoCode. I think this is a relevant problem to study today.

**Strengths And Weaknesses:**

Strengths:
I find the overall presentation of the paper to be very good. Core ideas are explained well and supported by relevant, interpretable experiments, ablations, and figures, while the appendix covers important details. I especially appreciated the extended discussion of “A Closer Look at Plotting Code as Alignment Signal” in the appendix. I think “plotting code as an alignment signal improves geometry solving” is an important (and interesting) claim of the paper, and the discussions in the main text plus the appendix offers a convincing argument against this being a spurious correlation. I also liked the idea of “textual debiasing” for geometry to cleanly study the unique role of understanding the pictorial problem description (as in rendered plot or the plotting code) in solving geometry problems. Compared to existing datasets, the addition of verifiably correct plotting code and increased problem diversity and complexity makes GeoCode valuable.
Weaknesses:
I find the description of textual debiasing in L256-296 is a bit vague as to how it’s exactly done. According to Table 6, it seems debiasing is sometimes helpful (depending on the alignment objective and on the dataset), though not universally. Are the other ways you can show how well it is working, e.g., how much textual debiasing truly removes the textual bias? E.g. How accurately can an LLM (no vision) solve the problems before vs after textual debiasing?
Minor weakness: The reasoning chain in GeoCode is taken as a matter of fact, but its exact format isn’t well-studied or varied. There could be room for improvement here.

---

> ### Author Rebuttal · Authors · 2026-03-30
>
> ## Overall Response
>
> We sincerely thank the reviewer for the highly positive evaluation and insightful questions. We are especially encouraged that the reviewer found the core idea of *plotting code as an alignment signal* convincing and well-supported.
>
> We will then provide our responses to each of the reviewer's weaknesses and questions.
> ## W1. Clarification of Textual Debiasing
>
> We thank the reviewer for pointing this out. We agree that the current description(L256–296) may appear insufficiently explicit and will revise it for clarity.
>
> In practice, textual debiasing is implemented via:
>
> 1. Carefully designed prompts with few-shot examples(App.47–51),
> 2. Structure-aware rewriting guided by plotting code.
>
> Specifically, we remove textual cues that can be directly inferred from the diagram(e.g. relative positions, equalities), while preserving all information necessary for solvability. Importantly, plotting code serves as a structural reference during this process, ensuring that essential geometric constraints are retained.
>
> We will update the main text to more clearly describe this procedure.
> ## W2 & Q2. Effectiveness of Debiasing and Role of Plotting Code
>
> To directly quantify both textual debiasing and the role of plotting code, we conduct addtionally controlled interventions using GPT-5:
> |Setting|Input|Accuracy|
> |---|---|---|
> |Original(text-only)|Text|74.25|
> |Original+Image|Text+Image|72.39|
> |Debiased(text-only)|Text|5.97|
> |Debiased+Image|Text+Image|42.16|
> |+Correct Code|Text+Image+Code|65.67|
> |+Wrong Code|Text+Image+Wrong Code|30.97|
>
> **(1) Effectiveness of textual debiasing.**
> Text-only accuracy drops sharply (74.25→5.97), indicating that shortcut-solvable signals are largely removed. Performance recovers when images are introduced(42.16), suggesting that solving now requires visual grounding. In contrast, original problems show minimal gap (74.25 vs 72.39), implying reliance on textual shortcuts.
>
> **(2) Role and correctness of plotting code.**
> Providing correct plotting code significantly improves performance(42.16→65.67), demonstrating that structured geometric representations effectively support reasoning. When incorrect code is provided, performance drops to 30.97, indicating that gains depend on correct geometric structure.
> ## Minor: Reasoning Chain Format
>
> We thank the reviewer. We agree that exploring more diverse or structured reasoning formats is an interesting direction, and will note this as a limitation and future work.
> ## Q1. Code-first vs Code-last vs Separate Prediction
>
> We additionally evaluate different training strategies using Qwen3-VL-8B:
>
> |Setting|Training Target|Accuracy|
> |---|---|---|
> |(a) Base|—|17.91|
> |(b) Code only|Code|20.15|
> |(c) Code → CoT → Answer|Code-first|26.49|
> |(d) CoT → Answer → Code|Code-last|23.88|
>
> **Findings:**
>
> (1) Predicting plotting code alone already yields consistent gains (+2.24), suggesting improved geometric perception.
>
> (2) Generation order matters: predicting code before reasoning achieves the best performance, indicating that it provides effective grounding.
>
> (3) Plotting code provides a distinct supervision signal: it is structured, explicitly encoding geometric entities and relations.
>
> **Conclusion.**
> Both *what to predict* (plotting code) and *when to predict it* (before reasoning) are important. We expect that scaling such supervision would further improve performance.
> ## Q3. Why 18K Problems?
>
> The current scale is primarily constrained by computational resources rather than an inherent limit of diversity::)
>
> The underlying symbolic space is extremely large. Prior work(e.g. GenesisGeo) demonstrates that millions of valid instances can be generated from DDAR predicate systems. This indicates that the available seed space is both large and diverse. Moreover, We do not observe saturation at this scale(diversity and performance continue to improve with more data in preliminary scaling experiments).
>
> Therefore, we have confidence that, with sufficient computational resources, our pipeline may be naturally scaled to produce significantly larger datasets.
> ## Q4. Comparison with Other Training Data
>
> A fully controlled comparison is challenging due to differences in supervision, scale, and training pipelines across datasets. Instead, we provide evidence from two perspectives:
>
> **(1) Data effectiveness.**
> Models trained on GeoCode consistently improve across multiple public benchmarks (Tab.3), demonstrating strong transferability. In preliminary experiments, training on existing datasets such as R-CoT, GeoSynth, and GeoTrust yields relatively limited improvements under comparable settings, suggesting that data quality and supervision form, rather than scale alone, play a critical role.
>
> **(2) Alignment mechanism.**
> Under a controlled setting (Tab.6), we vary only the supervision form. Caption-based supervision yields limited gains, while plotting-code supervision leads to substantially larger improvements.
>
> ---
> Typo in manuscript: Qwen3-VL-7B → Qwen3-VL-8B

---

> > ### Author Rebuttal · Reviewer_bYMk · 2026-04-03
> >
> > My concerns have been addressed, and I still feel good about this submission and maintain my accept rating.

---

### Decision · Program_Chairs · 2026-04-30

**Decision:**

Accept (regular)

**Comment:**

Summary of reviews:
* bYMk (A) praises the presentation and extended disucssion, as well as the proposed dataset. "I find the overall presentation of the paper to be very good. Core ideas are explained well and supported by relevant, interpretable experiments, ablations, and figures, while the appendix covers important details."
* 3aNp (WR / borderline) Initially praises the paper for "addressing a key hurdle within multimodal geometry reasoning" and showing the usefulness of the synthetic data for training and improvements on several benchmarks.  Critiques are insufficient evaluation of the complexity of the proposed dataset, over-reliance on LLM-based validation for dataset quality, and laock of comparison to other benchmarks. The rebuttal addresses these concerns, but the reviewer then says that the novelity is insufficient.  The authors provide an explanation of novelty.
* khDD (WA) says the approach is principled and well-validated and that "plotting code as an explicit alignment target is the most novel contribution", as well as that transfer to public benchmark is effective. The R critiques reliance on GPT-OSS-120B and notes some unexpected results that are not explained and a limitation of the dataset. Concerns are fully resolved in rebuttal.
* ALpY (WR) praises the presentation and experimental results but critiques the paper for limited novelty in data generation approach, potentially limited generality of approach, and unclear design decisions.  The main remaining concern is limited novelty.

Overall, the AC is in favor of accepting the paper, due to the agreed-on clear presentation, principled and well-validated method, and demonstration of effective transfer to the real benchmarks.  Based on these reviews and author responses, the AC believes that the paper has a clear and well-validated contribution, despite that it may relate to and build on previous approaches.